# Diagonal nematicity in the pseudogap phase of $HgBa_2CuO_{4+\delta}$

H. Murayama[1], Y. Sato [1], R. Kurihara[1], S. Kasahara [1], Y. Mizukami[2], Y. Kasahara[1], H. Uchiyama [3,4], A. Yamamoto[5], E.-G. Moon[6], J. Cai [7,8], J. Freyermuth[7,9], M. Greven[7], T. Shibauchi [2] & Y. Matsuda [1]

The pseudogap phenomenon in the cuprates is arguably the most mysterious puzzle in the field of high-temperature superconductivity. The tetragonal cuprate $HgBa_2CuO_{4+\delta}$, with only one $CuO_2$ layer per primitive cell, is an ideal system to tackle this puzzle. Here, we measure the magnetic susceptibility anisotropy within the $CuO_2$ plane with exceptionally high-precision magnetic torque experiments. Our key finding is that a distinct two-fold in-plane anisotropy sets in below the pseudogap temperature $T^*$, which provides thermodynamic evidence for a nematic phase transition with broken four-fold symmetry. Surprisingly, the nematic director orients along the diagonal direction of the $CuO_2$ square lattice, in sharp contrast to the bond nematicity along the Cu-O-Cu direction. Another remarkable feature is that the enhancement of the diagonal nematicity with decreasing temperature is suppressed around the temperature at which short-range charge-density-wave formation occurs. Our result suggests a competing relationship between diagonal nematic and charge-density-wave order in $HgBa_2CuO_{4+\delta}$.

---

[1] Department of Physics, Kyoto University, Kyoto 606-8502, Japan. [2] Department of Advanced Materials Science, University of Tokyo, Chiba 277-8561, Japan. [3] Materials Dynamics Laboratory, RIKEN SPring-8 Center, 1-1-1 Kouto, Sayo, Hyogo 679-5148, Japan. [4] Research and Utilization Division, Japan Synchrotron Radiation Research Institute (SPring-8/JASRI), 1-1-1 Kouto, Sayo, Hyogo 679-5198, Japan. [5] Graduate School of Engineering and Science, Shibaura Institute of Technology, 3-7-5 Toyosu, Koto-ku, Tokyo 135-8584, Japan. [6] Department of Physics, Korea Advanced Institute of Science and Technology, Daejeon 305-701, Korea. [7] School of Physics and Astronomy, University of Minnesota, Minneapolis, MN 55455, USA. [8] Present address: Physics Department, University of Maryland, College Park, MD 20742-4111, USA. [9] Present address: Department of Physics, The Ohio State University, Columbus, OH 43210-1117, USA. Correspondence and requests for materials should be addressed to Y.M. (email: matsuda@scphys.kyoto-u.ac.jp)

n hole-doped high transition temperature ($T_c$) cuprates, anomalous electronic states, including Fermi arcs, charge-density waves (CDWs) and $d$-wave superconductivity, emerge below the pseudogap onset temperature $T^{*}$ [1]. The pseudogap formation has been controversially discussed in terms of either a crossover phenomenon or a continuous phase transition. In the former scenario, the pseudogap represents a precursory gap of the $d$-wave superconductivity, and the phase fluctuations of pre-formed Cooper pairs destroy the superconducting order. In the latter scenario, the pseudogap emerges as a consequence of a spontaneous symmetry breaking. A continuous phase transition at $T^{*}$ is often argued to imply the presence of a quantum critical point, with associated fluctuations that may give rise to the high-$T_c$ superconductivity and strange-metal behaviours. Furthermore, the pseudogap order is thought to be intertwined (or compete) with other types of order, such as CDW order [2–7]. Until now, several types of broken symmetry, including broken translational, rotational, inversion and time-reversal symmetry, have been deduced from various experiments, including scanning tunnelling microscopy [8–13], polarised neutron scattering [14–18], polar Kerr [19], optical [20] and thermoelectric measurements [21,22]. Despite these tremendous efforts, the presence or absence of a continuous phase transition has been a highly controversial issue.

Recent torque magnetometry measurements of the anisotropic susceptibility within the $ab$ planes of YBa$_2$Cu$_3$O$_{6+\delta}$ (YBCO) revealed that the in-plane anisotropy displays a significant increase with a distinct cusp at $T^{*}$, consistent with the possible existence of a nematic phase transition [23]. However, in YBCO, the fourfold ($C_4$) rotational symmetry is already broken due to the orthorhombic crystal structure with one-dimensional (1D) CuO chains, and thus no further rotational symmetry breaking is expected. Moreover, in bilayer YBCO, the coupling of the CuO$_2$ planes in the unit cell may further affect the symmetry breaking [16]. Therefore, the investigation of an underdoped single-layer system with tetragonal symmetry, such as hole-doped HgBa$_2$CuO$_{4+\delta}$ (Hg1201), is essential to clarify whether a nematic phase transition is an intrinsic and universal property of the high-$T_c$ cuprates. Figure 1 displays the temperature-doping phase diagram of Hg1201. Similar to other hole-doped cuprates, the phase diagram contains CDW, superconducting and pseudogap regimes [6,7,17,18,24,25]. Short-range CDW order along [100]/[010], with wave vector close to 0.28 r.l.u., forms a dome-shaped boundary inside the pseudogap regime [6,7]. In zero magnetic field, CDW domains with typical size of a few nanometres appear.

Measurements of the magnetic torque $\boldsymbol{\tau} = \mu_0 V \mathbf{M} \times \mathbf{H}$ have a high sensitivity for the detection of magnetic anisotropy, where $\mu_0$ is space permeability, $V$ is the sample volume, and $\mathbf{M}$ is the magnetisation induced by external magnetic field $\mathbf{H}$ (Fig. 2a, see also Methods section). Torque is a thermodynamic observable that is equal to the derivative of the free energy with respect to angular displacement. Torque measurements performed for a range of directions of $\mathbf{H}$ within the tetragonal $ab$ plane of Hg1201 test whether or not the pseudogap state breaks the fourfold crystal symmetry. In this configuration, $\tau$ is a periodic function of twice the azimuthal angle $\phi$ measured from the $a$ axis:

$$\tau_{2\phi} = \frac{1}{2}\mu_0 H^2 V[(\chi_{aa} - \chi_{bb})\sin 2\phi - 2\chi_{ab}\cos 2\phi] \quad (1)$$

where $\chi_{ij}$ is the susceptibility tensor defined as $M_i = \sum_j \chi_{ij} H_j$ ($i,j = a, b, c$). For a tetragonally symmetric system, $\tau_{2\phi}$ should be zero because $\chi_{aa} = \chi_{bb}$ and $\chi_{ab} = 0$. Nonzero values of $\tau_{2\phi}$ appear when the tetragonal symmetry is broken by a new electronic or magnetic state; $C_4$ rotational symmetry breaking is revealed by $\chi_{aa} \neq \chi_{bb}$ and/or $\chi_{ab} \neq 0$. The former and the latter states are illustrated in Fig. 2b, c, where the $C_4$ symmetry breaking occurs along [100]/[010] direction (bond nematicity with $B_{1g}$-symmetry)

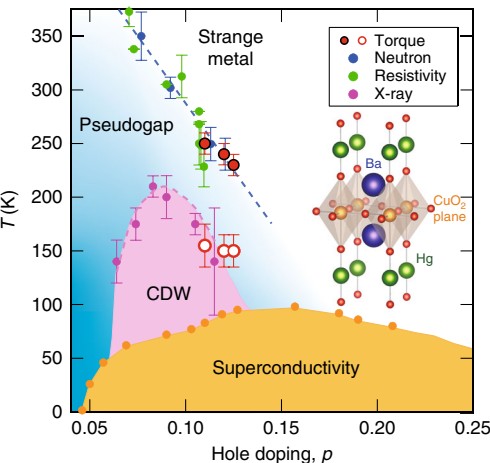

**Fig. 1** Temperature-doping phase diagram of Hg1201. Orange filled circles represent $T_c$ (see the Methods section). Blue and green filled circles represent the pseudogap onset temperature $T^{*}$ determined by neutron scattering [17,18] and resistivity measurements [24], respectively. Purple filled circles show the CDW onset temperature $T_{CDW}$ determined by resonant X-ray diffraction [6,7]. Red filled circles represent the onset temperature of diagonal nematicity determined by in-plane torque magnetometry, which lies on the pseudogap line (blue dashed line). Red open circles represent the temperature at which a suppression of the nematicity occurs, which is close to $T_{CDW}$. The inset shows the crystal structure of Hg1201

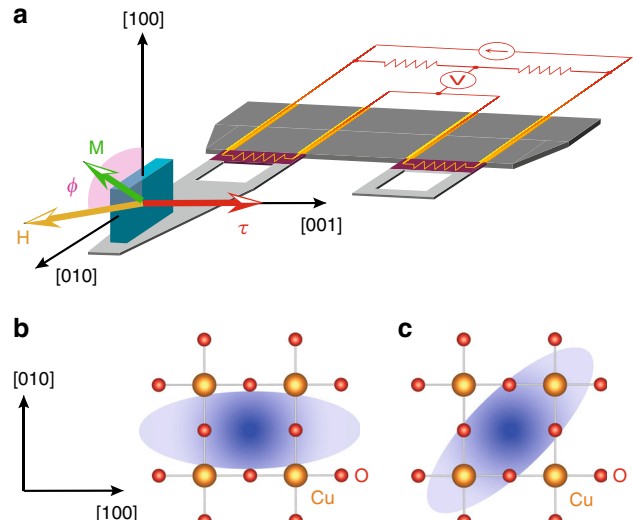

**Fig. 2** Torque magnetometry determination of nematic order. **a** The experimental configuration for in-plane torque magnetometry, by which magnetic torque $\boldsymbol{\tau} = \mu_0 V \mathbf{M} \times \mathbf{H}$ is measured. Yellow, green and red arrows indicate the directions of $\mathbf{H}$, $\mathbf{M}$ and $\boldsymbol{\tau}$, respectively. The magnetic field $\mathbf{H}$ is rotated within the tetragonal $ab$ plane. A single-crystalline sample of Hg1201 is mounted on the piezoresistive lever, which forms an electrical bridge circuit with the neighbouring reference lever. **b** Schematic picture of bond nematicity with $B_{1g}$ symmetry, where the nematicity appears along the Cu-O-Cu direction. For this nematicity, $\chi_{aa} \neq \chi_{bb}$ and $\chi_{ab} = 0$. **c** Diagonal nematicity with $B_{2g}$ symmetry, where the nematic director is along the diagonal direction of CuO$_2$ square lattice. For this nematicity, $\chi_{aa} = \chi_{bb}$ and $\chi_{ab} \neq 0$

and [110] direction (diagonal nematicity with $B_{2g}$-symmetry) of the CuO$_2$ plane.

Here, by measuring the in-plane magnetic susceptibility anisotropy with exceptionally precise torque magnetometry, we

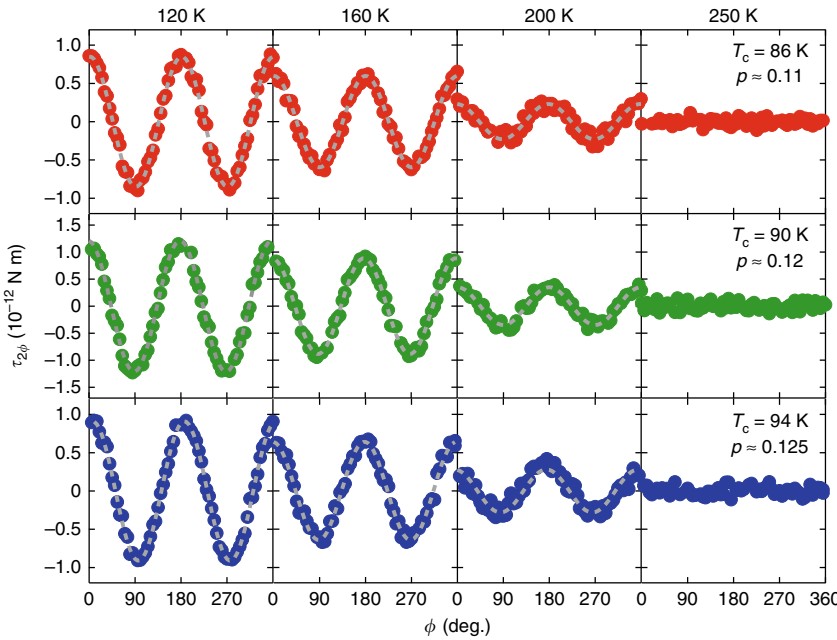

**Fig. 3** Twofold oscillations of magnetic torque in the $CuO_2$ planes. Upper panels show the torque curves $\tau_{2\phi}$ as a function of the azimuthal angle $\phi$ for $p \approx$ 0.11. Middle and lower panels show $\tau_{2\phi}$ for $p \approx$ 0.12 and 0.125, respectively

report thermodynamic signatures of a nematic phase transition at $T^*$ of Hg1201. Importantly, the observed twofold in-plane anisotropy below $T^*$ indicates that the nematic director in Hg1201 orients along the diagonal direction of the $CuO_2$ square lattice. Furthermore, the enhancement of the diagonal nematicity with decreasing temperature is suppressed around the temperature at which short-range CDW formation occurs. These results suggest a competing relationship between diagonal nematic and CDW order, shedding light on the pseudogap physics in cuprates.

## Results

**Magnetic torque**. The upper, middle and lower panels of Fig. 3 depict the magnetic torque curves measured as a function of $\phi$ for Hg1201 crystals with three different hole concentrations $p \approx 0.11$, 0.12 and 0.125 with $T_c = 86$, 90 and 94 K, respectively (see Supplementary Fig. 1 and Methods for sample characterisation). The approximate crystal sizes are $150 \times 140 \times 30\ \mu m^3$ ($p \approx 0.11$), $90 \times 90 \times 40\ \mu m^3$ ($p \approx 0.12$) and $120 \times 150 \times 50\ \mu m^3$ ($p \approx 0.125$). To avoid the mixing of out-of-plane components in the magnetic torque, the magnetic field ($|\mu_0 H| = 4$ T) is precisely applied in the $ab$ plane with out-of-plane misalignment less than $0.1°$ by controlling two superconducting magnets and a rotating stage situated at the top of the cryostat (see Supplementary Fig. 2, Supplementary Fig. 3 and Methods section). For all crystals, the twofold oscillation is absent at high temperatures, which is consistent with the tetragonal crystal symmetry. At low temperatures, however, the emergence of distinct twofold oscillations $\tau_{2\phi}$ is observed. This provides direct evidence for nematicity, indicating that the fourfold rotational symmetry at high temperature is broken down to the twofold rotational symmetry ($C_4 \rightarrow C_2$). Moreover, the twofold oscillation follows the functional form $\tau_{2\phi} = A_{2\phi} \cos 2\phi$, i.e., $\chi_{ab} \neq 0$ and $\chi_{aa} = \chi_{bb}$, which demonstrates the emergence of diagonal nematicity. This is in sharp contrast to the bond nematicity reported for double-layer $YBa_2Cu_3O_{6+\delta}$[21–23] and various iron-based superconductors[26], where the anisotropy axis is along the Cu–O–Cu and Fe–Fe directions.

**Temperature dependence of nematicity**. Figure 4a–c depicts the $T$ dependence of $2\chi_{ab}$ along with that of $\chi_{aa} - \chi_{bb}$ for $p \approx 0.11$,

0.12 and 0.125, respectively. While $\chi_{aa} - \chi_{bb}$ is almost temperature independent and negligibly small within the resolution, $2\chi_{ab}(T)$ shows characteristic temperature dependence. For all doping levels, $\chi_{ab} = 0$ at high temperatures. At the temperatures shown by solid arrows in Fig. 4a–c, $\chi_{ab}$ becomes finite and grows rapidly as the temperature is lowered. We plot these temperatures as red filled circles in Fig. 1. Obviously, the onset temperatures of the nematicity lie on the pseudogap line in the doping-temperature phase diagram determined by other methods, consistent with spontaneous macroscopic $C_4$ rotational symmetry breaking at $T^*$, i.e., with the notion that the onset of the pseudogap is characterised by a nematic phase transition.

As a natural consequence of the tetragonal crystal structure, the pseudogap phase with $C_2$ symmetry is expected to form domains with different preferred directions in the $ab$ plane. The observation of a nonzero $\chi_{ab}$ indicates a domain number imbalance. We note that although the magnitude of $\chi_{ab}$ for the three crystals is of the same order, $2\chi_{ab}$ for $p \approx 0.12$ is a few times larger than for $p \approx 0.11$ and 0.125. The torque curves remain unchanged for field-cooling conditions at different field angles. To check the influence of the strain on the side of the crystal attached to the cantilever, we measured the torque after remounting the crystal rotated by $90°$ (Supplementary Fig. 4a, b). The direction of the nematicity is unchanged relative to the crystal axes after the crystal rotation. Although the magnitude of $\chi_{ab}$ is enhanced, possibly due to a change in the imbalance of the number of the domains, the temperature dependence of $\chi_{ab}(T)$ is essentially the same in a wide temperature range below $T^*$ (Supplementary Fig. 4c). This implies that a large fraction of the domains are pinned by the underlying crystal conditions, such as internal stress, disorder and crystal shape, which may be consistent with the absence of hysteresis and thermal history effects. It also suggests that uniaxial pressure may reverse the direction of the nematicity. We note that a somewhat analogous situation is encountered in polar Kerr-effect measurements on YBCO, where the sign of the time-reversal symmetry breaking is fixed at temperatures significantly above $T^*$[19]. It has been reported that Hg1201 contains short-range oxygen chains[27,28]. We point out that these chains are not relevant to the observed

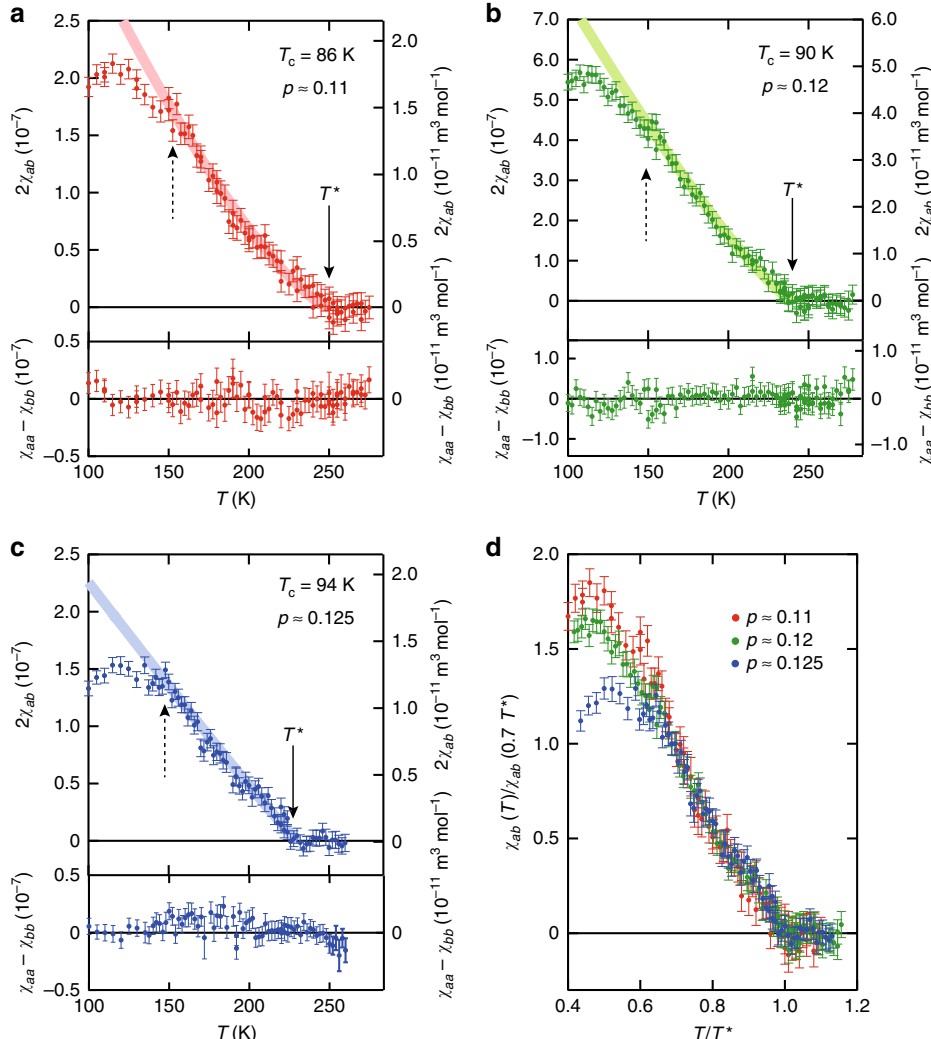

**Fig. 4** Anisotropy of magnetic susceptibility. **a** Temperature dependence of $2\chi_{ab}$ (upper panel) and $\chi_{aa} - \chi_{bb}$ (lower panel) for $p \approx 0.11$. **b**, **c** Same plots for $p \approx 0.12$ and 0.125, respectively. Solid arrows indicate the onset temperatures of $\chi_{ab}$, which well coincides with $T^*$, as shown in Fig. 1. Dashed arrows indicate the temperatures at which $\chi_{ab}$ deviates from the extrapolation from high temperature shown by the bold lines. These temperatures are close to $T_{CDW}$, as shown in Fig. 1. **d** $\chi_{ab}$, normalised by the values at $T/T^* = 0.7$ for different doping levels plotted as a function of $T/T^*$. The data collapse onto a universal curve, except for the low-temperature regime where a suppression of the $\chi_{ab}$ is observed, indicating a scaling behaviour. Error bar represents s.d. of the sinusoidal fit to the $\tau_{2\phi}(\phi)$ curves

nematicity, because the direction of the chains is along [100]/[010], while the direction of the nematicity is along [110]. Moreover, the chains are quite stable and form at high temperature, well above $T^*$, whereas the nematicity clearly onsets at $T^*$.

**Comparison with the bond nematicity in YBCO**. We now compare the diagonal nematicity of Hg1201 with that of the bond nematicity of YBCO. In Fig. 4d, the scaled values $\chi_{ab}(T)/\chi_{ab}(0.7T^*)$ for $p \approx 0.11$, 0.12 and 0.125 are presented. The three curves collapse onto the same curve within the error bars down to $T \sim 0.65T^*$. Although measurements in a wider doping range would be required to establish universal scaling behaviour in Hg1201, we note that scaling behaviour as well as the characteristic super-linear temperature dependence have also been reported for YBCO in a wide doping range[23]. In orthorhombic YBCO, the amplitude of the nematicity can be analysed in terms of excess nematicity defined as $\Delta\eta(T) \equiv \eta(T) - \eta(T^*)$, where $\eta \equiv (\chi_{aa} - \chi_{bb})/(\chi_{aa} + \chi_{bb})$. It has been shown that $\eta(T)$ is temperature independent above $T^*$. In a rather wide doping

range, $\Delta\eta(T) \sim 5 \times 10^{-3}$ at $T \sim T^*/2$[23]. In Hg1201, $2\chi_{ab}$ is $\sim 2 - 5 \times 10^{-7}$ at $T \sim T^*/2$. Using the in-plane magnetic susceptibility $\chi_{aa} = \chi_{bb} \sim 1 - 2 \times 10^{-5}$ (see the Methods section), we obtain $2\chi_{ab}/(\chi_{aa} + \chi_{bb}) \sim 1 - 5 \times 10^{-2}$. Thus the amplitude of the diagonal nematicity in Hg1201 is comparable with that of the bond nematicity in YBCO. These results suggest that, despite the different nematic directions, the diagonal nematicity in Hg1201 resembles the bond nematicity in YBCO.

**Suppression of diagonal nematicity below $T_{CDW}$**. Deep inside the nematic phase, at temperatures well below $T^*$, $\chi_{ab}(T)$ exhibits another anomaly. For $p \approx 0.125$, $\chi_{ab}(T)$ is strongly suppressed below $T \sim 150$ K (Fig. 4c). Although this anomaly is less pronounced for $p \approx 0.11$ and 0.12, $\chi_{ab}(T)$ shows a deviation from an extrapolation from higher temperatures at $T = 140 - 160$ K, as shown by red and green lines in Fig. 4a, b. The observed suppression of the nematicity well below $T^*$ is in stark contrast to YBCO, in which the nematicity grows monotonically with decreasing temperature without discernible anomaly down to $T_c$. The suppression temperatures of the diagonal nematicity

observed in Hg1201 are close to the CDW transition temperature $T_{CDW}$ determined by resonant X-ray diffraction measurements[6,7]. It has been shown that below this temperature, the Hall constant levels off and the planar resistivity exhibits $T^2$ dependence[24]. The CDW modulation is along [100]/[010], which is the same direction as the bond nematicity in YBCO, but differs from the diagonal nematicity in Hg1201. The fact that $\chi_{aa} - \chi_{bb}(T)$ remains negligibly small below $T_{CDW}$ implies that the CDW domain size is much smaller than the sample dimensions, which results in the cancellation of the $\sin2\phi$ oscillations with opposite signs from different domains. This is consistent with the scattering experiments, which revealed CDW correlations of a few nm[6,7].

## Discussion

It should be stressed that the nature of the nematicity in Hg1201 differs from that in YBCO in several regards. First, in the case of Hg1201, purely spontaneous $C_4 \rightarrow C_2$ rotational symmetry breaking of the electron system occurs in Hg1201 with a tetragonal crystal symmetry. Second, the diagonal nematicity with $B_{2g}$ symmetry in Hg1201 is in stark contrast to the bond nematicity with $B_{1g}$ symmetry in YBCO. Third, while the growth of the diagonal nematicity is suppressed by the CDW formation in Hg1201, no discernible anomaly is observed at $T_{CDW}$ in YBCO.

Our results shed light on the pseudogap physics. Nematicity in the pseudogap phase is universal regardless of the number of CuO$_2$ layers per primitive cell. Moreover, the diagonal nematicity in Hg1201, observed in this study for the first time, is associated with the $B_{2g}$ representation, in sharp contrast to the CDW order along the bond direction ($B_{1g}$), which demonstrates that the nematicity is not a precursor of the CDW. Notice that in other cuprates such as YBCO and Bi$_2$Sr$_2$CaCu$_2$O$_{8+\delta}$, nematicity and CDW develop along the same bond direction and the precursor issue has not been clear.

The super-linear temperature behaviour of the nematicity in YBCO and Hg1201 suggests that the pseudogap transition may not belong to the two- or three-dimensional Ising universality class. One possible explanation is that the nematicity may not be a primary order parameter, whereas another possibility is that pseudogap phenomena are associated with physics beyond the Landau paradigm.

Next, we discuss possible explanations of the diagonal nematicity. Our results suggest that the diagonal nematicity is unlikely due to a Pomeranchuk instability of the Fermi surface, which prefers symmetry breaking along the bond direction[29]. We note that we cannot rule out the possibility of complete rotational symmetry breaking ($C_4 \rightarrow C_1$). The pattern of an intra-unit-cell loop-current order with $C_1$ symmetry may be consistent with the diagonal direction[30]. However, the interpretation of the polarised neutron results[14–18] in terms of loop-current-order is not unique, and the observed magnetism appears to be dynamic rather than truly static. In addition, it is an interesting open question whether the loop-current order would explain the bond nematicity in bilayer cuprates such as YBCO. Recent resistivity measurements on tetragonal La$_{2-x}$Sr$_x$CuO$_4$ thin films report broken $C_4$ symmetry even above $T^*$[31]. Moreover, the direction of the nematicity is not fixed by the crystal axes, and depends on temperature and hole concentration. At the present stage, the relationship between this transport work and our thermodynamic result is an open question.

More scenarios have been suggested recently. One is based on octupolar order of Cu-3$d$ orbital for bond nematicity under the Landau paradigm[32], another is based on percolation of local pseudogaps which may be consistent with nematic domain phenomena near $T^*$[33], and a third is based on phenomena beyond the Landau paradigm, such as doped quantum spin liquid, which

may explain the superlinear onset intrinsically[34]. Further theoretical and experimental investigations are highly desired to pin down a mechanism of the diagonal nematicity.

## Methods

**Materials.** We have studied single crystals of Hg1201 grown by two different techniques. For $p \approx 0.11$ and 0.125, small single crystals were grown by a solid-state reaction method[35,36]. For $p \approx 0.12$, a small crystal was cut from a large crystal grown by a flux method[37]. The hole concentration for $p \approx 0.12$ was controlled by annealing the crystal at high temperatures under oxygen atmosphere. The hole-doping levels were determined from the superconducting transition temperature $T_c$ by magnetisation measurements[36]. Supplementary Fig. 1a–c shows the temperature dependence of the normalised magnetisation measured for $p \approx 0.11$, 0.12 and 0.125, respectively. The crystals exhibit sharp superconducting transitions with $T_c$ of 86, 90 and 94 K for $p \approx 0.11$, 0.12 and 0.125, respectively. The directions of the crystalline axes were determined by X-ray diffraction measurements.

**Torque magnetometry.** Magnetic torque was measured by the piezoresistive micro-cantilever technique[23,38,39]. With a tiny amount of instant glue, tiny single crystals of Hg1201 were carefully mounted onto the piezoresistive lever, which forms an electrical bridge circuit with the neighbouring reference lever. Then, the magnetic torque $\tau$ is measured as $\tau = \frac{at^2}{2\pi_L} \frac{\Delta R}{R_s}$, where $a$ is the leg width, $t$ is the leg thickness, $\pi_L$ is the piezoresistive coefficient, and $R_s$ is the resistance of the cantilever[40]. For precise measurements of the in-plane magnetic torque, we use a system consisting of a 2D vector magnet and a mechanical rotator (Supplementary Fig. 2), which enables us to rotate the magnetic field **H** within the crystal $ab$ plane. It should be noted that we use two coordinate systems. One is the $XYZ$-system based on the rotation mechanism, which we use to determine the position of the sample plane (Supplementary Fig. 2). The other is the $a$, $b$, $c$-axes of the sample. To avoid confusion, we use uppercase letters for the polar ($\Theta$) and azimuthal ($\Phi$) angles for the former system, and lowercase ($\theta$ and $\phi$) for the latter.

In the experiments, we first determine the position of the sample plane by measuring the out-of-plane torque as a function of angle $\Theta$ from the $z$-axis (Supplementary Fig. 2). The measurements of $\tau(\Theta)$ curves are repeated at various $\Phi$. Figure 3a shows typical curves of $\tau(\Theta)$ for $p \approx 0.11$ at 80 K in the superconducting state. At this temperature, $\tau(\Theta)$ exhibits a sharp change when **H** crosses the $ab$ plane. The data are fitted by a symmetric polynomial function, and accordingly, we precisely determine the alignment of the $ab$ plane, $\Theta_{ab}$, at which $\tau(\Theta) = 0$. Supplementary Fig. 3b shows $\Theta_{ab}$, as a function of angle $\Phi$ for $p \approx 0.11$. Via computer control of the vector field and mechanical rotator systems, we eliminate the misalignment $\Delta\Theta_m$ at each $\Phi$. Then we rotate **H** within the $ab$ plane with a field misalignment better than 0.1°, without changing setup or removing the sample. Measurements of $\tau(\phi)$ curves are repeated at all the temperatures in Fig. 4. By analysing all the twofold oscillation curves, we obtain the temperature dependences of $\chi_{ab}$ and $\chi_{aa} - \chi_{bb}$. After completing all the experiments, the sample alignment is double checked at 80 K to confirm that the sample did not move during the measurement.

Supplementary Fig. 5a, b shows typical out-of-plane anisotropy $\Delta\chi_\perp = \chi_{cc} - \chi_{aa}$ in the normal state ($T > T_c$) for $p \approx 0.11$ and 0.125, respectively, obtained by rotating **H** within a plane that includes the $c$-axis. The $\tau(\theta)$ curves exhibit purely paramagnetic response with no discernible hysteresis components, and are well fitted by

$$\tau_{2\theta}(\theta, T, H) = \frac{1}{2}\mu_0 H^2 V \Delta\chi_\perp \sin2\theta, \qquad (2)$$

which yields $\pi$ periodic oscillations with respect to the $\theta$ rotation. Here, $\theta$ is the polar angle from the $c$-axis, $\Delta\chi_\perp = \chi_{cc} - \chi_{aa}$ is the difference between the $c$-axis and the in-plane susceptibilities. From the amplitude of the $\tau(\theta)$ curves, the temperature dependences of $\Delta\chi_\perp$ are obtained. At high temperatures, the magnitude of $\Delta\chi_\perp$ decreases nearly linearly with temperature, while the data deviate downward from this approximate behaviour below $T^*$, which represents the pseudogap formation[23].

We estimate $\chi_{aa}$ from the magnetic susceptibility of a powder sample, $\chi_{powder} = 2/3\chi_{aa} + 1/3\chi_{cc}$, reported for optimally doped Hg1201[41], and the out-of-plane anisotropy, $\Delta\chi_\perp = \chi_{cc} - \chi_{aa}$ (Supplementary Fig. 5). The estimated $\chi_{aa}$ is $\sim 1 - 2 \times 10^{-5}$.

**Simulation of magnetic torque.** To examine the presence of in-plane anisotropy in Hg1201, we have simulated the amplitude of the torque for the cases with/without in-plane anisotropy. In Hg1201, the off-diagonal components of the magnetic susceptibility tensor are given as $\chi_{ab} = \chi_{ba}$ and $\chi_{ac} = \chi_{ca} = \chi_{bc} = \chi_{cb} = 0$. In this case, the magnetic torque $\boldsymbol{\tau} = (\tau_a, \tau_b, \tau_c)$ can be written as

$$\tau_a = \mu_0 V[\chi_{ab} H_a H_c - (\chi_{cc} - \chi_{bb})H_b H_c] \qquad (3)$$

$$\tau_b = \mu_0 V[\chi_{ab} H_b H_c - (\chi_{cc} - \chi_{aa})H_a H_c] \qquad (4)$$

$$\tau_c = \mu_0 V[\chi_{ab}(H_b^2 - H_a^2) + (\chi_{aa} - \chi_{bb})H_a H_b] \qquad (5)$$

where **H** = ($H_a$, $H_b$, $H_c$) is the magnetic field.

To evaluate the contribution due to the misalignment between the sample and the cantilever (mount misalignment), we introduce $xyz$ coordinates for the cantilever, where the bending direction of the lever is in the $xy$ plane (Supplementary Fig. 6a). The signal of the cantilever is only sensitive to the torque along the $z$-axis, because the bending of the lever is limited to the $xy$ plane due to its structure. Signal detected by the lever is $\tau^{lever} = \boldsymbol{\tau} \cdot \mathbf{e}_z = \tau_a e_a + \tau_b e_b + \tau_c e_c$, where $\mathbf{e}_z = (e_a, e_b, e_c)$ is the normal unit vector for the bending $(xy)$ plane. Thus, when the sample plane is perfectly aligned with the lever, i.e., $\mathbf{e}_z = (0, 0, 1)$, only in-plane torque $\tau_c$ is measured. Of course, in reality, we have unavoidable mount misalignment of the sample with respect to the lever. However, the in-plane anisotropy can be accurately measured under the aligned field condition $\mathbf{H} = (H_a, H_b, 0)$, because $\boldsymbol{\tau} = (0, 0, \tau_c)$ even if mount misalignment is present. In practice, there also exists a small difference between the $xy$ plane of the lever and the $XY$ plane of the rotating stage, which cannot be determined accurately. However, we emphasise that this difference only results in a slight modification of $\mathbf{e}_z$ and does not alter our discussion when the magnetic field is correctly applied within the $ab$ plane.

Next, we discuss the effect of the misalignment of the magnetic field with respect to the sample plane (field misalignment) as a mixing of the out-of-plane torque components occurs only when both mount misalignment and field misalignment are present. Although this field misalignment can be eliminated in our experiment by using 2D vector magnet ($\theta_0 < 0.1°$), here we hypothetically assume that the magnetic field is applied in a plane with a much larger misalignment of $\theta_0 = 5°$, which is tilted in an arbitrary $\phi_0$ direction (Supplementary Fig. 6b). The field angle from the $ab$ plane is then given by $\Delta\theta = \theta_0 \cos(\phi - \phi_0)$. Below we calculate the expected torque response, $\tau^{lever}$, as a function of the field misalignment $(\theta, \phi)$ for the following cases assuming that the mount misalignment of the sample with respect to the lever is as large as the apparent misalignment from the $XY$ plane found at 80 K (Supplementary Fig. 3b, $\Theta_m = -11.9°$ and $\Phi_0 = 109°$).

(A) The system preserves the fourfold rotational symmetry, i.e., $\chi_{ab} = 0$. Thus, when we rotate the magnetic field, only the out-of-plane component of the magnetic torque contributes to the signal (Supplementary Fig. 6c, d).

(B) The system breaks rotational symmetry, i.e., $\chi_{ab} \neq 0$. By using $\chi_{ab}$ shown in Fig. 4, we calculate the torque response for the ideal case without mount misalignment between the sample and the cantilever (Supplementary Fig. 6e, f).

(C) The system breaks rotational symmetry, i.e., $\chi_{ab} \neq 0$. We also include an apparent mount misalignment of the sample with respect to the lever (Supplementary Fig. 6g, h).

The colour plots in Supplementary Fig. 6c, e, g show the expected amplitude of torque, $\tau^{lever}$, as a function of field misalignment $(\theta, \phi)$ for $p \approx 0.11$ at $T = 180$ K. Solid lines show the trajectories of the magnetic field in a plane with given misalignment. Supplementary Fig. 6d, f, h demonstrates the angular dependence of the magnetic torque, $\tau_{2\phi}(\phi)$, expected for the assumed misaligned planes of the magnetic field. For comparison, we also show $\tau_{2\phi}(\phi)$ when the magnetic field is exactly applied within the $ab$ plane (black lines, $\theta_0 = 0$ and $\phi_0 = 0$).

In case (A), when the field is applied within the $ab$ plane, twofold oscillations are absent even if the sample is mounted on the lever with non-zero mount misalignment. If the magnetic field is applied within a misaligned plane, twofold oscillations due to the out-of-plane anisotropy would appear. However, we note that the phase of the oscillations is unrelated to the crystal axes in this case. One may accidentally observe twofold oscillations. However, we emphasise that we repeated our measurements on several different samples with different mountings. Therefore, such accidental oscillations would not explain the reproducibility of the diagonal nematicity, which is always observed along the [110] direction of the crystals. It should be also noted that the amplitude of the accidental twofold oscillations is much smaller than the observed signal. In the calculation, we use a large field misalignment, much larger than the actual values of our experiments. With the actual setup of $\theta_0 < 0.1°$, the oscillations shown in Supplementary Fig. 6d become negligibly small. Therefore, both the amplitude and the phase due to the field misalignment, even if they exist, are inconsistent with the experimental observations.

In case (B), the twofold oscillations are not influenced by the misalignment of $5°$ of the magnetic field. This is because only the magnetic torque along the $z$-axis is probed in the experiments as the cantilever bends only within the $xy$ plane.

In case (C), the phase and amplitude of the twofold oscillations are modified from the oscillations without mount misalignment, because both in-plane anisotropy and misalignments induced out-of-plane components contribute to the signal. In this case, it is expected that the phase of the twofold oscillations changes with temperature, because the relative weight of the two components changes with temperature. In our experiments, however, the phase of the twofold oscillations is always fixed as $\tau_{2\phi} \sim \cos 2\phi$ and does not change with temperature.

In addition, we demonstrate in Supplementary Fig. 7, the expected changes in the temperature dependence of the magnetic susceptibility anisotropies for case (C), when both mount misalignment and field misalignment are present. We note that even for a large field misalignment of $\theta_0 = 5°$, the contribution from the out-of-plane component only appears as a slight shift of the original signal, whereas the onset of $2\chi_{ab}$ is clearly observed. This confirms that our experimental results intrinsically represent the in-plane diagonal nematicity which onsets below $T^*$. It

should also be noted that when we have field misalignment, a non-zero component of $\chi_{aa} - \chi_{bb}$, would appear. This should give rise to a phase shift of the twofold oscillations, while it is never observed in the experiments.

**Torque magnetometry under finite out-of-plane field**. To further confirm the validity of our experiments and demonstrate that issues related to the field misalignment are not the reason for the diagonal nematicity, we measured the magnetic torque in the presence of a non-zero out-of-plane magnetic-field component.

Supplementary Fig. 8a, b shows the magnetic torque for $p \approx 0.125$ recorded under conical field rotations at non-zero $\theta$ for $T = 240$ K and 160 K, respectively. In Supplementary Fig. 9a–c, we map the results of the magnetic torque in the $(\theta, \phi)$ plane at $T = 240$, 180 and 160 K, respectively. Supplementary Fig. 9d–f depicts the expected torque amplitude when both field and mount misalignments are present. For these calculations of the expected torque, the apparent mount misalignment of $\Theta_0 = 10.6°$ and $\Phi_0 = 274°$ determined by the out-of-plane torque measurements is used. In Supplementary Fig. 9g, we plot the expected torque, in case the direction of the sample plane is misidentified. Here, an inclined plane of $\theta' = (\theta - 5°) \cos(\phi - 20°)$ is assumed as a misidentified direction of the sample plane.

As we see in Supplementary Figs 8a and 9a, in-plane anisotropy is absent at $T = 240$ K ($> T^*$) and the torque amplitude shows symmetric behaviour in the $(\theta, \phi)$ plane. This indicates that $\theta = 90°$ correctly captures the direction of the $ab$ plane of the crystal. Otherwise, the response of the torque will be distorted in the $(\theta, \phi)$ plane as depicted in Supplementary Fig. 9g. The emergence of the in-plane anisotropy below $T^*$ is clearly seen at $\theta = 90°$. As a result, the torque response becomes asymmetric in the $(\theta, \phi)$ plane. It should be noted that this deformation of the torque amplitude in the $(\theta, \phi)$ plane below $T^*$ is essentially different from the simple distortion expected for the case when the sample plane is misidentified. The excellent agreement of the observed torque amplitude with the expected response strongly supports that neither field misalignment nor mount misalignment is the reason for the observed onset of the twofold oscillations, which represents the emergence of the diagonal nematicity below $T^*$.

## Data availability
The data that support the findings of this study are available on request from the corresponding author.

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

## Acknowledgements
We thank A. Fujimori, T. Hanaguri, S. Kivelson, H. Kontani, P.A. Lee, D. Pelc, S. Sachdev, L. Taillefer, T. Tohyama, C. Varma, H. Yamase, Y. Yanase and G. Yu for fruitful discussions. This work was supported by Grants-in-Aid for Scientific Research (KAKENHI) (Nos. JP25220710, JP15H02106, JP15H03688, JP16K13837, JP18H01177 and JP18H05227) and on Innovative Areas "Topological Material Science" (No. JP15H05852) and "Quantum Liquid Crystals" (No. JP19H05824) from Japan Society for the Promotion of Science (JSPS). This work was partly performed using facilities of the Institute for Solid State Physics, the University of Tokyo. H.U. thanks Alfred Baron for making the work possible in Materials Dynamics laboratory, RIKEN SPring-8 Center. The work at the University of Minnesota was funded by the Department of Energy through the University of Minnesota Center for Quantum Materials under DE-SC-0016371. E.-G.M. acknowledges the financial supports from the POSCO Science Fellowship of POSCO TJ Park Foundation and NRF of Korea under Grant no. 2017R1C1B2009176.

## Author contributions
H.U., A.Y., J.C., J.F. and M.G. grew the high-quality single-crystalline samples. H.M., Y.S., R.K. and S.K. performed the magnetic torque measurements. H.M. and Y. Mizukami performed the X-ray diffraction measurements. H.M., Y.S., S.K., Y.K., E.-G.M. and Y. Matsuda analysed the data. H.M., S.K., E.-G.M., T.S., M.G. and Y. Matsuda discussed and interpreted the results and prepared the paper.

## Additional information

**Competing interests:** The authors declare no competing interests.

