## [Peer Review File · Nature Communications]

Reviewers' comments:

Reviewer #1 (Remarks to the Author):

H. Murayama and co-workers have used a macroscopic technique of torque magnetometry to investigate the existence of an anisotropic susceptibility in a family of high- T_c superconductors, $\text{HgBa}_{2-x}\text{CuO}_{4+d}$ (Hg1201), as a signature of nematic order within the 'pseudogap' phase. In this tetragonal system spin susceptibility is expected to be equal when a magnetic field is applied along tetragonal a or b axis (i.e. $\chi_{aa} = \chi_{bb}$), while diagonal susceptibility is expected to be zero (i.e. $\chi_{ab} = 0$). They used small-volume crystals to reveal a two-fold symmetry of torque as a magnetic field, applied parallel to the CuO_2 plane, is rotated around the 4-fold axis (c axis), with maximum appearing along [110] direction. This 2-fold symmetry along with its appearance in the normal state above T_c but below some characteristic 'pseudogap' temperature is taken as an evidence of a nematic phase in Hg1201. This is undoubtedly an intriguing observation in the context of theoretical paradigm of intertwined order parameters in such correlated systems. Although authors presented data showing that C_4 symmetry is reduced well above T_c they need to address the following points.

(1) As the authors have pointed out, a phase with C_2 -symmetry would inevitably form domains below the 'pseudogap' temperature, if there is a transition. In order for a net torque to exist there has to be an imbalance in the number of these symmetry-related domains. How did the authors surmise that in $p \sim 0.12$ sample the size of these domains were tens of microns implying a total domain population of the order of a hundred? Is there a numerical or theoretical justification for such large domains to form? Are the authors ruling out an imbalance is from a much larger population of smaller domains to generate equivalent anisotropy in susceptibility?

(2) It is known that patches of ordered oxygen chains form in Hg1201. These oxygen chains are highly anisotropic, forming with a varying degrees of domain-size distribution as a function of doping. While global symmetry of a crystal may remain tetragonal, locally though, these chains reduce structural symmetry to C_2 . This may be due to lattice strain and distortions as well as their population imbalance, especially in smaller crystals. It would be good to address this issue that presence of such chains do not alter their primary findings.

(3) AF ordered phase depicted in the x-T phase diagram and its reference in the text are inconsistent with each other. If the authors do not have any evidence of an AF order then it might be better to restrict the discussion to an experimentally known phase diagram for Hg1201.

(4) Authors noted in the methods section that SC transitions are sharp in these crystals. They should provide a figure (e.g. in supplementary section) showing SC transitions and explicitly quote the SC transition widths of the specific crystals that they have used in their torque measurements.

(5) Compositions chosen are close enough to each other that no meaningful differences in CDW transition temperatures identified as a temperature where χ_{ab} deviates from higher temperature behavior is apparent. The assignment of T^* as well as its width are not very convincing (see Fig. S4). The spread of composition is not large enough to claim that scaling behavior, $\chi_{ab}(T)/\chi_{ab}(0.7T^*)$, is universal. In this context, it would have been rather beneficial to have a larger spread in their choice of compositions.

(6) These measurements were done with an applied field of 4 Tesla, a choice that was not discussed or motivated. Given a quadratic dependence on magnetic field one would expect observability of torque to be significantly better in stronger fields, which would be quite important at higher temperatures. Have the authors looked into any influence on temperature dependence, especially T^* , on field strength used?

Reviewer #2 (Remarks to the Author):

Dear Editor

The manuscript presents torque measurements on three doping compositions of Hg1201 cuprates in the underdoped regime -- overlapping with the doping region where CDW and quantum oscillations have been observed. The measurement and the analysis follows closely on the heels of the similar (<https://arxiv.org/pdf/1706.05214.pdf>) measurement in YBCO cuprates by an overlapping team of scientists. In this contexts its motivation and significance is to to validate and extend the universality of the sharp thermodynamic anomaly across pseudogap boundary cuprates into a broader class of cuprate materials.

My main concern relates the validity of the data and some confusion the reader will encounter in the data presentation.

1) Fig.3 (or its caption) does not indicate the field at which torque measurements (indicated only in the text). . In particular $\sim 1\text{pJ}$ torque @4T appears to be very small signal not only in the physical units (10^{-6} bohr magnetons per Cu ion ?) but also for the comparizon with the magnitude of possible experimental biases. The main experimental artefact and an alternative source of sine2theta component in torque would be the "wobbling of the sample" -- i.e., imperfect alignment of the plane of rotation with crystallographic ab-plane. Since the measured effect is so small and the uniaxial (χ_c vs $\chi_{a,b}$) anisotropy of magnetic susceptibility in cuprates is well known, it would help if authors provide necessary information regarding the upper limit for the expected $\tau_2\phi$ component due to wobbling of a-b plane. In particular, the rotation mechanism of the rotator probe may have temperature dependence.

2) The units of χ in the vertical axis of Fig 4 are not specified -- it would help a lot to understand the magnitude of the effect if authors specified the units in emu/mol or bohr magnetons per Tesla per Cu ion or some such -- at least in the caption or in the main text. Comparison of the magnitude of anisotropy with $\chi_a(T)$ or $\chi_c(T)$ would also illuminate the data.

3) More then half of the text of the paper attempts a broad overview of experimental situation not directly relevant to understanding of the data presented in the paper. It would help the broad readership of this Communication if discussion is more focused on the data and reader is referred to reviews for broader context. In particular it would help if the magnitude of the observed effect (about 10^{-6} bohr magnetons per Cu ion if I am interpreting correctly the data in Figure 3) is discussed in the context of thermodynamic behavior of heat capacity. Is the effect so small because it is greatly reduced by "nematic" domains ? is there a hysteresis associated with these domains ? Is it small to begin with ? etc.

I would wholeheartedly recommend publication of this important data set after issues outlined above are addressed.

Thank you very much for sending us the two referee reports. We would like to thank both reviewers for their insightful comments, which led us to improve our manuscript. Both reviewers acknowledge the significance of our results. Reviewer #1 states that this is undoubtedly an intriguing observation and Reviewer #2 wholeheartedly recommends publication after his/her concerns are addressed. Below we reply to the comments by the referees. Having addressed all the issues raised by both reviewers, we trust that you will find the revised manuscript suitable for publication in *Nature Communications*.

Reviewers' comments:

Reviewer #1 (Remarks to the Author):

H. Murayama and co-workers have used a macroscopic technique of torque magnetometry to investigate the existence of an anisotropic susceptibility in a family of high-T_c superconductors, HgBa₂CuO_{4+d} (Hg1201), as a signature of nematic order within the 'pseudogap' phase. In this tetragonal system spin susceptibility is expected to be equal when a magnetic field is applied along tetragonal a or b axis (i.e. $\chi_{aa}=\chi_{bb}$), while diagonal susceptibility is expected to be zero (i.e. $\chi_{ab}=0$). They used small-volume crystals to reveal a two-fold symmetry of torque as a magnetic field, applied parallel to the CuO₂ plane, is rotated around the 4-fold axis (c axis), with maximum appearing along [110] direction. This 2-fold symmetry along with its appearance in the normal state above T_c but below some characteristic 'pseudogap' temperature is taken as an evidence of a nematic phase in Hg1201. This is undoubtedly an intriguing observation in the context of theoretical paradigm of intertwined order parameters in such correlated systems. Although authors presented data showing that C4 symmetry is reduced well above T_c they need to address the following points.

We gratefully acknowledge Reviewer #1 for carefully reading our manuscript. We also thank Reviewer #1 for several important comments. In the following, we respond to his/her comments.

(1) As the authors have pointed out, a phase with C2-symmetry would inevitably form domains below the 'pseudogap' temperature, if there is a transition. In order for a net torque to exist there has to be an imbalance in the number of these symmetry-related

domains. How did the authors surmise that in $p \sim 0.12$ sample the size of these domains were tens of microns implying a total domain population of the order of a hundred? Is there a numerical or theoretical justification for such large domains to form? Are the authors ruling out an imbalance is from a much larger population of smaller domains to generate equivalent anisotropy in susceptibility?

The measurements before and after the sample is rotated by 90 degrees shown in Fig.S4a-c demonstrate that the nematic direction is coupled to the crystal shape (which is somewhat irregular), and this can explain the domain number imbalance. However, the estimate of domain size from the fact that signal strengths are similar in different samples is somewhat crude. There is no theoretical justification for such large domains, and as suggested by Reviewer #1, we cannot rule out the possibility of an imbalance from a much larger population of smaller domains (especially in the c -direction). In the revised manuscript, we have eliminated the discussion of the estimate of the domain size.

(2) It is known that patches of ordered oxygen chains form in Hg1201. These oxygen chains are highly anisotropic, forming with a varying degrees of domain-size distribution as a function of doping. While global symmetry of a crystal may remain tetragonal, locally though, these chains reduce structural symmetry to C_2 . This may be due to lattice strain and distortions as well as their population imbalance, especially in smaller crystals. It would be good to address this issue that presence of such chains do not alter their primary findings.

This is an important comment. The presence of the local random oxygen chains pointed out by Reviewer #1 are not relevant to the observed nematicity because of the following reasons:

- (i) The direction of the chains is along $[100]/[010]$ while the direction of the nematicity is along $[110]$. We see no reason that the chains should induce the $[110]$ diagonal nematicity.
- (ii) The chains are quite stable and form at high temperature, well above T^* , whereas the nematicity clearly onsets at T^* .
- (iii) The relatively small spatial scale associated with the chains would imply that a significant imbalance of chain directions is required to detect their effect by torque magnetometry, which unlikely occurs for different samples from different batches, especially given the low density of the local random chains.

We have added a relevant citation and the above discussion in the revised manuscript.

(3) AF ordered phase depicted in the x-T phase diagram and its reference in the text are inconsistent with each other. If the authors do not have any evidence of an AF order then it might be better to restrict the discussion to an experimentally known phase diagram for Hg1201.

We agree with Reviewer #1. To restrict the discussion to an experimentally observed phase diagram, we eliminate the AFM order regime in the phase diagram (Fig. 1) as suggested.

(4) Authors noted in the methods section that SC transitions are sharp in these crystals. They should provide a figure (e.g. in supplementary section) showing SC transitions and explicitly quote the SC transition widths of the specific crystals that they have used in their torque measurements.

In the supplementary section, we have added magnetic susceptibility data for the samples used for the torque measurements.

(5) Compositions chosen are close enough to each other that no meaningful differences in CDW transition temperatures identified as a temperature where χ_{ab} deviates from higher temperature behavior is apparent. The assignment of T^* as well as its width are not very convincing (see Fig. S4). The spread of composition is not large enough to claim that scaling behavior, $\chi_{ab}(T)/\chi_{ab}(0.7T^*)$, is universal. In this context, it would have been rather beneficial to have a larger spread in their choice of compositions.

We agree that measurements in wider doping regime would be desirable. Unfortunately, however, it is difficult to extend our measurements to a wider doping range for the following reasons. T^* in the heavily underdoped regime is close to or higher than room temperature. In our setup, it is dangerous to heat up the samples close to room temperature because the glue used to fix the crystal to cantilever would begin to melt. Moreover, to obtain overdoped crystals, we attempted anneals in high oxygen pressure, but uniformly oxygen-doped crystals were not obtained. In the revised manuscript, we toned down the scaling discussion and mention “Although measurements in a wider doping regime would be required to establish universal scaling behaviour in Hg1201, we note that scaling behaviour as well as the characteristic super-linear temperature

dependence have also been reported in YBCO in a wide doing range.”

(6) These measurements were done with an applied field of 4 Tesla, a choice that was not discussed or motivated. Given a quadratic dependence on magnetic field one would expect observability of torque to be significantly better in stronger fields, which would be quite important at higher temperatures. Have the authors looked into any influence on temperature dependence, especially T^* , on field strength used?

In the present measurements, the magnetic field is very accurately rotated within the *ab*-plane. By using a vector magnet and mechanical rotator system, we attained the misalignment from the *ab*-plane less than 0.1 degree. The highest field, which is 4 T, is limited by the maximum field of the vector magnet. As the torque signal is proportional to H^2 in the paramagnetic state, we chose the maximum available field. We have not measured the field dependence of the nematic transition, but it is well established that T^* is not influenced by a magnetic field of 4 T.

Reviewer #2 (Remarks to the Author):

Dear Editor

The manuscript presents torque measurements on three doping compositions of Hg1201 cuprates in the underdoped regime -- overlapping with the doping region where CDW and quantum oscillations have been observed. The measurement and the analysis follows closely on the heels of the similar (<https://arxiv.org/pdf/1706.05214.pdf>) measurement in YBCO cuprates by an overlapping team of scientists. In this contexts its motivation and significance is to validate and extend the universality of the sharp thermodynamic anomaly across pseudogap boundary cuprates into a broader class of cuprate materials.

We thank Reviewer #2 for the close reading of our manuscript and for the comments and suggestions.

My main concern relates the validity of the data and some confusion the reader will encounter in the data presentation.

1) Fig.3 (or its caption) does not indicate the field at which torque measurements (indicated only in the text). . In particular $\sim 1\text{pJ}$ torque @4T appears to be very small signal not only in the physical units (10^{-6} bohr magnetons per Cu ion ?) but also for the comparizon with the magnitude of possible experimental biases. The main experimental artefact and an alternative source of $\sin 2\theta$ component in torque would be the "wobbling of the sample" -- i.e., imperfect alignment of the plane of rotation with crystallographic ab -plane. Since the measured effect is so small and the uniaxial (χ_c vs $\chi_{a,b}$) anisotropy of magnetic susceptibility in cuprates is well known, it would help if authors provide necessary information regarding the upper limit for the expected τ_{ϕ} component due to wobbling of a - b plane. In particular, the rotation mechanism of the rotator probe may have temperature dependence.

We have added the applied magnetic field in Fig. 3, as suggested by Reviewer #2. As for the susceptibility unit, we use the dimensionless SI unit for volume susceptibility ($\chi=M/H$), not Bohr magnetons per Cu, which is a unit of magnetization M . We point out that a typical χ_{aa} value for the cuprates is 10^{-5} (dimensionless) per volume, which gives 10^{-4} Bohr magneton per Cu at 1 T. What we measure is an off-diagonal component χ_{ab} as large as a few percent of χ_{aa} , which is not very small. Indeed, a similar magnitude of the anisotropy $\chi_{aa} - \chi_{bb}$ has been reported for YBCO.

As discussed in the SI, the magnetic field can be rotated within the ab -plane with a misalignment less than 0.1 degree by using a vector magnet. The twofold oscillations due to the misalignment θ_m is calculated as $\Delta\chi^m = (\chi_a - \chi_c) \sin 2\theta_m$. $\Delta\chi^m$ is negligibly small even at $\theta = 1$ degree. The wobbling suggested by Reviewer #1 is negligibly small, which is confirmed by the absence of twofold oscillations above T^* . The mechanical rotator is situated at the top of the cryostat, so it is always at room temperature. The misalignment also has been checked in the superconducting state at low temperatures (see Fig.S3). Thus the observed temperature dependence cannot be explained by the misalignment effect.

2) The units of χ in the vertical axis of Fig 4 are not specified -- it would help a lot to understand the magnitude of the effect if authors specified the units in emu/mol or bohr magnetons per Tesla per Cu ion or some such -- at least in the caption or in the main text. Comparison of the magnitude of anisotropy with $\chi_a(T)$ or $\chi_c(T)$ would also illuminate the data.

As mentioned above, in the SI units that we used, the magnetic susceptibility per

volume is dimensionless. At the suggestion by Reviewer #2, we added the unit m^3/mol in the right axis in Fig.4a-c. There is no measurement of χ_{aa} on Hg1201 single crystals. We then estimate the in-plane χ_{aa} from the magnetic susceptibility of a powder sample, $\chi_{\text{pow}} = 2/3 \chi_{aa} + 1/3 \chi_{cc}$, reported for optimally-doped Hg1201 Ref. [37], and the out-of-plane anisotropy, $\Delta\chi_{\text{perp}} = \chi_{cc} - \chi_{aa}$ (Fig. S5). We obtained $\chi_{aa} \sim 1-2 \times 10^{-5}$. We mentioned this in the Methods.

3) More than half of the text of the paper attempts a broad overview of experimental situation not directly relevant to understanding of the data presented in the paper. It would help the broad readership of this Communication if discussion is more focused on the data and reader is referred to reviews for broader context. In particular it would help if the magnitude of the observed effect (about 10^{-6} bohr magnetons per Cu ion if I am interpreting correctly the data in Figure 3) is discussed in the context of thermodynamic behavior of heat capacity. Is the effect so small because it is greatly reduced by "nematic" domains? Is there a hysteresis associated with these domains? Is it small to begin with? etc.

In the revised manuscript, we compare the T -dependence and amplitude of the nematicity of Hg1201 with those of YBCO. We mention that, despite the different nematic directions, the diagonal nematicity in Hg1201 resembles the bond nematicity in YBCO.

Given that the origin of the nematicity is not understood, it is hard to estimate how the specific heat changes at T^* from the nematicity found in the magnetic torque. Until now, a specific heat anomaly has not been reported. Detecting a tiny change of the specific heat at relatively high temperatures, where the lattice contribution overwhelms the electronic contribution, is a very interesting but challenging issue.

I would wholeheartedly recommend publication of this important data set after issues outlined above are addressed.

Reviewers' comments:

Reviewer #1 (Remarks to the Author):

Authors have addressed all issues and made changes to the manuscript accordingly. It is clear that there is an anisotropy appearing far above T_c in at least three different samples, which is of general interest. I recommend it for publication.

Reviewer #2 (Remarks to the Author):

The issue is that the in-plane magnetic anisotropy reported in this work (and in the related cuprate compound, YBCO [6]) is very small. Quite generally, torque measurement have a number of biases that need to be eliminated. However, the magnitude of the effect claimed in this work requires much far more stringent analysis of biases in the experimental torque setup.

In particular, there is a large ($\sim 2E-6$ between 100K and 200K) onset of uniaxial magnetic anisotropy ($\chi_c - \chi_a$) (Fig S5), which -- coupled with sample misalignment -- could alone be responsible for the reported in-plane anisotropy signal. The elaborate alignment setup described in the manuscript is however insufficient to assure required level of alignment (less the ~ 1 deg) over entire temperature range. The reason for that is that there are 3 different planes that all need to be aligned below certain limit before the observed behavior in Fig 4 can be attributed to the onset of magnetic anisotropy in the ab-crystal plane (rather than to the onset of in the uniaxial magnetic anisotropy ($\chi_c - \chi_a$)) :

- (i) the crystallographic ab-plane,
- (ii) the plane in which lever bends and
- (iii) the plane of rotation of magnetic field or the stage on which lever is mounted.

The misalignment of planes (i) and (ii) -- orientation of the sample on the lever after it is (glued or greased) -- can result from both vertical tilt of the sample around axis parallel to the long axis of the lever as well as from a slight rotation of the sample around the axis perpendicular to the lever (vertical direction in Fig 2A). Manuscript does not provide details of how sample was attached and how it was aligned on the lever -- I will assume that it was attached by hand as it is usually done and therefore I will assume that the misalignment of the sample on the lever (plane (i) with respect to plane (ii)) to be about 5 degrees, or about 0.1 radian. At one particular temperature one can adjust nonzero "wobble" of the rotation stage (or magnetic field) -- plane (iii) -- to cancel the "wobble" of the ab-plane of sample -- plane (i) -- as it bends the lever. This adjustment, I presume, is done at one temperature only (the manuscript is not clear about this so far) and therefore the cancellation is not balanced once one of the parameters involved ($\chi_c - \chi_a$ -- uniaxial anisotropy) changes its value as temperature is increased or decreased (Current manuscript is unclear about what is the temperature at which the magnet alignment is made: Fig S3 suggests 80K whereas Fig 3 suggests that subtraction/alignment was made close to room temperature because in [6] the ab-plane anisotropy is finite even above T^* (see Fig 2 inn [6])). In summary, it appears that the 0.1deg alignment claimed in Methods is a result of cancellation at one temperature of misalignment between plane (i) and (ii) by adjusting the angle between planes (ii) and (iii). It would help if authors could comment on that. Such cancellation can always be achieved for small wobble angles because lever always picks component of torque perpendicular to the plane of bending.

This might suggest that the observed onset of in-plane anisotropy below T^* (Fig 4) is a direct result of onset of the decrease in the uniaxial ($\chi_a - \chi_c$) anisotropy (Fig S5) at the same temperature, rather than an onset of in-plane anisotropy. The magnitude of the observed planar anisotropy, $\sim 0.2E-6$ between 100 and 200K in Fig 4a is close to expected magnitude due to

wobbling $\sim 0.1(\chi_c - \chi_a)$ where the factor 0.1 is wobbling angle (5 deg) and $(\chi_c - \chi_a)$ changes by $\sim 2E-6$ in the same temperature range (Fig S5a).

Few more comments and requests.

1) Please provide mass (or volume or molar content) of each sample discussed in the paper. If unavailable, please provide details of calibration of the bridge circuit, and details of how the absolute values of χ (in Figs 3 and 4) has been obtained from measured bridge voltage.

3) how is sample attached on the lever -- grease? epoxy ? How is the sample aligned on the lever ? any information on the degree of misalignment as glue/grease expands/contracts with temperature ?

4) Has alignment test been repeated at several temperatures? Is there a way to assess the degree to which alignment is maintained over entire measured temperature range? How stable is the rotation mechanism over a broad temperature range ?

5) Figure 3 shows evolution of the two-fold rotation anisotropy with temperature -- the amplitude of the $\sin 2\theta / \cos 2\theta$ is a direct indication of inferred anisotropy at a given temperature. However, it appears that the plots in Fig 4 were obtained in a different way -- by doing the temperature scan at fixed angle. If so, the temperature dependence in Fig 4 will also pick the changes over a broad temperature range in the bending stiffness of the lever as well as temperature dependence of the bridge circuit on the lever.

What is the upper bound on these effects ? Has these been considered in the current Figure 4 and accompanying discussion?

5) In response to my point 2 (in the previous response) the authors added estimate of $\chi_{aa} \sim 1-2 \times 10^{-5}$ and added corresponding text in the manuscript. This estimate seems to be off by an order of magnitude (see [3]).

They also -- incorrectly -- point out that χ_{aa} of $1E-5$ corresponds to $1E-4 \mu_B/T$ --- it instead corresponds to $1.5E-3 \mu_B/T$ -- an order of magnitude off [4].

6) In response to my point 1 Authors write "As discussed in the SI, the magnetic field can be rotated within the ab-plane with a misalignment less than 0.1 degree by using a vector magnet. The twofold oscillations due to the misalignment θ_m is calculated as $\Delta\chi_m = (\chi_a - \chi_c) \sin 2\theta_m$. $\Delta\chi_m$ is negligibly small even at $\theta = 1$ degree. The wobbling suggested by Reviewer #1 is negligibly small, which is confirmed by the absence of twofold oscillations above T^* . The mechanical rotator is situated at the top of the cryostat, so it is always at room temperature. The misalignment also has been checked in the superconducting state at low temperatures (see Fig.S3). Thus the observed temperature dependence cannot be explained by the misalignment effect."

As discussed above, the validity of their interpretation requires alignment of three planes (i),(ii),(iii) -- adjusting vector magnet at one temperature only achieves cancellation (at one temperature) between wobbling of plane (i) with respect to (ii) and plane (ii) with respect to (iii). This cancellation is destroyed as soon as temperature changes. This concern is even more severe because the magnitude of the expected signal due to wobbling is about 1/10th of the change in $\Delta\chi_{\perp} = \chi_{cc} - \chi_{aa}$, -- close to the observed magnitude. Here 0.1 is the angle of misalignment of the sample on the lever (plane (i) with respect to plane (ii)) in radians. In this context, the measured signal is direct consequence of the temperature behavior of $\chi_c - \chi_a$ rather than in-plane anisotropy $\chi_{bb} - \chi_{aa}$.

[1] Y. Itoh and T. Machi, arxiv:0804.0911

[2] $a = 3.886 \text{ \AA}$ and $c = 9.517 \text{ \AA}$, from Zhao et al, DOI:10.1002/adma.200600931

[3] I dont have Ref 37 readily available (few readers will). Fig 6 in [1] gives 1.2×10^{-4} emu/mole at 200K which is about 1.4×10^{-6} emu/cm³ for HgBa₂CuO₄ (molar volume of HgBa₂CuO₄ is 86.5 (cm³) / mol, see [2]). The value of $\chi_{aa} \sim 1 - 2 \times 10^{-5}$ is off by an order of magnitude since the value for χ_{powder} in [1] is 1.4×10^{-6} emu/cm³ and the value of $\chi_{\text{perp}} = \chi_{\text{cc}} - \chi_{\text{aa}}$ in Fig S5 is $\sim 4.5 \times 10^{-6}$ emu/cm³ at 200K; according to authors, $\chi_{aa} = \chi_{\text{powder}} - 1/3 \Delta\chi_{\perp}$. ($\chi_{\text{powder}} = 2/3 \chi_{aa} + 1/3 \chi_{\text{cc}}$, $\Delta\chi_{\perp} = \chi_{\text{cc}} - \chi_{aa}$)

[4] 1×10^{-5} emu/cm³ = 1.5×10^{-3} μ_{B}/T for HgBa₂CuO₄ (molar volume of HgBa₂CuO₄ is 86.5 (cm³) / mol -- see [2]).

[5]

[6] Y. Sato et. al. Nat. Phys. 13 1076 (2017)

Reviewer #1 (Remarks to the Author):

Authors have addressed all issues and made changes to the manuscript accordingly. It is clear that there is an anisotropy appearing far above T_c in at least three different samples, which is of general interest. I recommend it for publication.

(Our reply)

We thank Reviewer #1 for her/his recommendation for publication of our paper in *Nat. Commun.*

=====
Reviewer #2 (Remarks to the Author):

(Reviewer's comment #2-0)

The issue is that the in-plane magnetic anisotropy reported in this work (and in the related cuprate compound, YBCO [6]) is very small. Quite generally, torque measurement have a number of biases that need to be eliminated. However, the magnitude of the effect claimed in this work requires much far more stringent analysis of biases in the experimental torque setup.

In particular, there is a large ($\sim 2E-6$ between 100K and 200K) onset of uniaxial magnetic anisotropy ($\chi_c - \chi_a$) (Fig S5), which -- coupled with sample misalignment -- could alone be responsible for the reported in-plane anisotropy signal. The elaborate alignment setup described in the manuscript is however insufficient to assure required level of alignment (less than ~ 1 deg) over entire temperature range. The reason for that is that there are 3 different planes that all need to be aligned below certain limit before the observed behavior in Fig 4 can be attributed to the onset of magnetic anisotropy in the ab-crystal plane (rather than to the onset of in the uniaxial magnetic anisotropy ($\chi_c - \chi_a$)) :

- (i) the crystallographic ab-plane,
- (ii) the plane in which lever bends and
- (iii) the plane of rotation of magnetic field or the stage on which lever is mounted.

The misalignment of planes (i) and (ii) -- orientation of the sample on the lever after it is (glued or greased) -- can result from both vertical tilt of the sample around axis parallel to the long axis of the lever as well as from a slight rotation of the sample around the axis perpendicular to the lever (vertical direction in Fig 2A). Manuscript does not provide

details of how sample was attached and how it was aligned on the lever -- I will assume that it was attached by hand as it is usually done and therefore I will assume that the misalignment of the sample on the lever (plane (i) with respect to plane (ii)) to be about 5 degrees, or about 0.1 radian. At one particular temperature one can adjust nonzero "wobble" of the rotation stage (or magnetic field) -- plane (iii) -- to cancel the "wobble" of the ab-plane of sample -- plane (i) -- as it bends the lever. This adjustment, I presume, is done at one temperature only (the manuscript is not clear about this so far) and therefore the cancellation is not balanced once one of the parameters involved ($\chi_c - \chi_a$ -- uniaxial anisotropy) changes its value as temperature is increased or decreased (Current manuscript is unclear about what is the temperature at which the magnet alignment is made: Fig S3 suggests 80K whereas Fig 3 suggests that subtraction/alignment was made close to room temperature because in [6] the ab-plane anisotropy is finite even above T^* (see Fig 2 in [6])).

In summary, it appears that the 0.1deg alignment claimed in Methods is a result of cancellation at one temperature of misalignment between plane (i) and (ii) by adjusting the angle between planes (ii) and (iii). It would help if authors could comment on that. Such cancellation can always be achieved for small wobble angles because lever always picks component of torque perpendicular to the plane of bending.

This might suggest that the observed onset of in-plane anisotropy below T^* (Fig 4) is a direct result of onset of the decrease in the uniaxial ($\chi_a - \chi_c$) anisotropy (Fig S5) at the same temperature, rather than an onset of in-plane anisotropy. The magnitude of the observed planar anisotropy, $\sim 0.2E-6$ between 100 and 200K in Fig 4a is close to expected magnitude due to wobbling $\sim 0.1(\chi_c - \chi_a)$ where the factor 0.1 is wobbling angle (5 deg) and ($\chi_c - \chi_a$) changes by $\sim 2E-6$ in the same temperature range (Fig S5a).

(Our reply #2-0)

We thank Reviewer #2 for taking her/his time for the second review of our manuscript. The main concern raised by Reviewer #2 is that the observed in-plane anisotropy below T^* may come from the misalignments of the sample with respect to the magnetic field and the cantilever. Here, we demonstrate that such misalignments are unable to explain the present results. To show this clearly, we have made new calculations on the torque response for the cases with/without diagonal nematicity, including the misalignment between the sample and the magnetic field /cantilever. As we describe the details below, our calculations rule out the misalignment as a source of two-fold oscillations below T^* .

In Hg1201, the off-diagonal components of the magnetic susceptibility tensor are given as $\chi_{ab} = \chi_{ba}$ and $\chi_{ac} = \chi_{ca} = \chi_{bc} = \chi_{cb} = 0$. In this case, magnetic torque $\boldsymbol{\tau} = (\tau_a, \tau_b, \tau_c)$ can be written as

$$\begin{aligned}\tau_a &= \frac{1}{2}\mu_0 V [\chi_{ab} H_a H_c - (\chi_{cc} - \chi_{bb}) H_b H_c] \\ \tau_b &= -\frac{1}{2}\mu_0 V [\chi_{ab} H_b H_c - (\chi_{cc} - \chi_{aa}) H_a H_c] \\ \tau_c &= \frac{1}{2}\mu_0 V [\chi_{ab} (H_b^2 - H_a^2) + (\chi_{aa} - \chi_{bb}) H_a H_b]\end{aligned}$$

where $\mathbf{H} = (H_a, H_b, H_c)$ is the magnetic field.

First, to evaluate the contribution of the misalignment between the sample and the cantilever, we introduce xyz coordinate for the cantilever, where the bending direction of the lever is in the xy plane (Fig. R1a). The signal of the cantilever is only sensitive to the torque along the z axis because the bending of the lever is limited to the xy plane due to its structure. Signal detected by the lever is $\tau^{\text{lever}} = \boldsymbol{\tau} \cdot \mathbf{e}_z = \tau_a e_a + \tau_b e_b + \tau_c e_c$, where $\mathbf{e}_z = (e_a, e_b, e_c)$ is the normal unit vector of the bending plane. Thus, when the sample plane is perfectly aligned with the lever, i.e. $\mathbf{e}_z = (0, 0, 1)$, only in-plane torque τ_c is measured. In practice, however, we have unavoidable misalignment between the lever and the sample plane, as pointed out by Reviewer #2, and thus we expect to observe the oscillations in the torque signal due to the out-of-plane anisotropy.

Next we include the misalignment of the magnetic field with respect to the sample plane. Although this misalignment in our experiments is negligibly small ($\theta_0 < 0.1$ deg), here we hypothetically assume that the magnetic field is applied in a plane with a much larger misalignment of $\theta_0 = 5$ deg, which is tilted in an arbitrary ϕ_0 direction (Fig. R1b). The field angle from the ab plane is then given by $\Delta\theta = \theta_0 \cos(\phi - \phi_0)$. This misalignment between the field and the sample will cause a mixing of the torque proportional to the out-of-plane anisotropy, $\chi_{cc} - \chi_{aa}$, shown in Fig.S5. Below we calculate the expected torque response τ^{lever} for the following cases assuming that the misalignment of the sample with respect to the lever is as large as the apparent misalignment found at 80 K (Fig. S3b, $\Delta\theta_m = -11.9$ deg and $\phi_0 = 109$ deg).

- (A) The system preserves the four-fold rotational symmetry, i.e. $\chi_{ab} = 0$. Thus, when we rotate the magnetic field, only the out-of-plane component of the magnetic torque contributes to the signal because of the misalignment (Fig.R1c and d). Even if we assume large misalignments between the sample and the field/cantilever, the

signal is quite small.

- (B) The system breaks rotational symmetry, i.e. $\chi_{ab} \neq 0$. By using χ_{ab} shown in Fig. 4, we calculate torque response for an ideal case without misalignment between the sample and the cantilever (Fig.R1e and f). The two-fold oscillations appear owing to the C_2 symmetry of the system.
- (C) The system breaks rotational symmetry, i.e. $\chi_{ab} \neq 0$. We also include a misalignment of the sample with respect to the lever (Fig.R1g and h).

Fig. R1: Expected magnetic torque for the cases with/without in-plane anisotropy.

The colour plots in Fig.R1c,e,g show the amplitude of torque, $\tau_{2\phi}^{\text{lever}}$, as a function of (ϕ, θ) for $\rho \approx 0.11$ at $T = 180 \text{ K}$. Solid lines show the trajectories of the magnetic field in a

plane with given misalignment. Fig.R1d, f, h demonstrate the angular dependence of the magnetic torque, $\tau_{2\phi}^{\text{lever}}(\phi)$, expected for the assumed misaligned planes. For comparison, we also show $\tau_{2\phi}^{\text{lever}}(\phi)$ when the magnetic field is exactly applied within the *ab*-plane (black lines, $\theta_0 = 0$ and $\phi_0 = 0$).

In case (A), when the field is applied within the *ab*-plane, two-fold oscillations are absent even if the sample is mounted on the lever with misalignment. If the magnetic field is applied within a misaligned plane, two-fold oscillations due to the out-of-plane anisotropy would appear. However, we note that the phase of the oscillations is unrelated to the crystal axes in this case. One may accidentally observe two-fold oscillations. However, it should be stressed that our experiments are repeated on several different samples with different mountings. Therefore, such accidental oscillations would not explain the reproducibility of the diagonal nematicity, which is always observed along the [110] direction of the crystals. It should be also noted that the amplitude of the two-fold oscillations is much smaller than the observed signal. In the calculation, we use a large misalignment θ_0 , much larger than the actual values of our experiments. With the actual set-up of $\theta_0 < 0.1$ deg, the oscillations shown in Fig.R1d become negligibly small. Therefore, both the amplitude and the phase due to misalignment, even if they exist, are inconsistent with the experimental observations.

In case (B), the two-fold oscillations are not influenced by the misalignment of 5 deg of the magnetic field. This is because only the magnetic torque along the z-axis is probed in the experiments as the cantilever bends only within the *xy*-plane.

In case (C), the phase and amplitude of the two-fold oscillations are modified from the oscillations without misalignments, because both in-plane anisotropy and misalignment induced out-of-plane component contribute to the signal. In this case, it is expected that the phase of the two-fold oscillations changes with temperature because relative weight of the two components changes with temperature. In our experiments, however, the phase of the two-fold oscillations is always fixed as $\tau_{2\phi} \sim \cos 2\phi$ and does not change with temperature.

In addition, we demonstrate in Fig.R2 the expected changes in the temperature dependence of the magnetic susceptibility anisotropies for the case (C) when we add the misalignments. We note that even for a large misalignment of $\theta_0 = 5$ deg, the contribution from the out-of-plane component only appears as a slight shift of the original signal, whereas the onset of $2\chi_{ab}$ is observed clearly. This confirms that our experimental results intrinsically represent the in-plane diagonal nematicity which onset below T^* . It should be also noted that when we have nonzero misalignment between the sample and the field, finite component of $\chi_{cc} - \chi_{aa}$ would appear. This should give rise to

a phase shift of the two- fold oscillations, while it is never observed in the experiments.

Fig. R2: Expected temperature dependence of magnetic susceptibility anisotropies for $p \approx 0.11$ for a magnetic field rotated in a misaligned plane.

In summary, we demonstrate that the Reviewer's concerns are unwarranted regarding the effects of misalignment of

- (i) the crystallographic ab-plane,
- (ii) the plane in which lever bends and
- (iii) the plane of rotation of magnetic field or the stage on which lever is mounted.

In the revised manuscript, we have added the above discussion, which we trust further strengthens our conclusions.

We now address the Reviewer's comment:

"Fig S3 suggests 80K whereas Fig 3 suggests that subtraction/alignment was made close to room temperature because in [6] the ab-plane anisotropy is finite even above T^* (see Fig 2 inn [6])."

In our previous work on YBCO, a nonzero in-plane anisotropy is present at $T > T^*$ because four-fold (C_4) rotational symmetry is already broken due to the orthorhombic crystal structure. Therefore, the investigation of a tetragonal system such as Hg1201 is definitely important to clarify the nematic phase transition at T^* . The alignment of the sample is determined at 80 K in the superconducting state by the out-of-plane magnetic torque -- we do not determine the alignment to eliminate the two-fold oscillations in the in-plane torque.

Once the alignment is determined, we repeat the measurements of $\tau(2\phi)$ curves at all the temperatures shown in Fig. 4. Consequently, in Hg1201, we find that in-plane anisotropy is absent at $T > T^*$, which is a natural result of the tetragonal crystal structure.

Few more comments and requests.

(Reviewer's comment #2-1)

Please provide mass (or volume or molar content) of each sample discussed in the paper. If unavailable, please provide details of calibration of the bridge circuit, and details of how the absolute values of χ (in Figs 3 and 4) has been obtained from measured bridge voltage.

(Our reply #2-1)

The volumes of the samples had been listed in the original manuscript (See the second paragraph of p.4.) In the revised manuscript, the procedure how to determine the absolute values of χ is included in the Methods section.

(Reviewer's comment #2-2)

how is sample attached on the lever -- grease? epoxy ? How is the sample aligned on the lever ? any information on the degree of misalignment as glue/grease expands/contracts with temperature ?

(Our reply #2-2)

A tiny amount of instant glue is used to fix the sample onto the lever. Since the sample is mounted by hand, misalignment between the sample and the lever is unavoidable. However, since the magnetic field is applied within the ab-plane, the misalignment between the sample and the lever is not important because τ_a and τ_b are always zero in this case.

We also note that the alignment of the sample does not change over the temperature range we have measured. If the sample moves with temperature, the phase of the two-fold oscillations should shift. The absence of such a phase shift in our experiments confirms that the sample does not move for the entire temperature range. In addition, in order to confirm that the sample did not move from the original position during the measurements, we always checked the alignment of the sample at 80 K after completing the experiments.

(Reviewer's comment #2-3)

Has alignment test been repeated at several temperatures? Is there a way to assess the degree to which alignment is maintained over entire measured temperature range? How stable is the rotation mechanism over a broad temperature range ?

(Our reply #2-3)

The misalignment of the sample ($\Delta\theta_m$) is determined at 80 K in the superconducting state (Fig. S3). As mentioned above, the absence of phase shift in two-fold oscillations confirms that the sample does not move for the entire temperature range.

A mechanical rotator with no backlash is equipped at the top of the cryostat; this rotator is always at room temperature, as noted in our previous response. Then the whole sample probe, which is in a variable temperature insert, is rotated around the z-axis. Therefore, the rotation mechanism is completely stable over a broad temperature range.

(Reviewer's comment #2-4)

Figure 3 shows evolution of the two-fold rotation anisotropy with temperature -- the amplitude of the $\sin 2\theta/\cos 2\theta$ is a direct indication of inferred anisotropy at a given temperature. However, it appears that the plots in Fig 4 were obtained in a different way -- by doing the temperature scan at fixed angle. If so, the temperature dependence in Fig 4 will also pick the changes over a broad temperature range in the bending stiffness of the lever as well as temperature dependence of the bridge circuit on the lever.

What is the upper bound on these effects ? Has these been considered in the current Figure 4 and accompanying discussion?

(Our reply #2-4)

In contrast to the Reviewer's speculation, we measured the magnetic torque as a function of azimuthal angle ϕ at ALL temperatures. By analyzing all the two-fold oscillations, we obtain the temperature dependences of the susceptibility anisotropies. This procedure is very important to eliminate the influence of the temperature dependence of the bridge circuit, and to evaluate the absolute values of the $\chi_{aa}-\chi_{bb}$ and χ_{ab} with exceptional precision.

(Reviewer's comment #2-5)

In response to my point 2 (in the previous response) the authors added estimate of $\chi_{aa} \sim 1-2 \times 10^{-5}$ and added corresponding text in the manuscript. This estimate seems to be off by an order of magnitude (see [3]).

They also -- incorrectly -- point out that χ_{aa} of $1E-5$ corresponds to $1E-4 \mu_B/T$ --- it instead corresponds to $1.5E-3 \mu_B/T$ -- an order of magnitude off [4].

(Our reply #2-5)

We believe that Reviewer #2 misunderstood the UNITS of the magnetic susceptibility. As explicitly commented in our previous reply, we use dimensionless volume susceptibility in SI units. To clarify this point, we show in Fig.R3 the magnetic susceptibility on powdered sample in SI and CGS units.

Fig. R3: Magnetic susceptibility on powdered sample of Hg1201 (data obtained from [37], same as Y. Itoh and T. Machi, arxiv:0804.0911).

(Reviewer's comment #2-6)

In response to my point 1 Authors write "As discussed in the SI, the magnetic field can be rotated within the ab-plane with a misalignment less than 0.1 degree by using a vector magnet. The twofold oscillations due to the misalignment θ_m is calculated as $\Delta\chi_m = (\chi_a - \chi_c) \sin 2\theta_m$. $\Delta\chi_m$ is negligibly small even at $\theta = 1$ degree. The wobbling suggested by Reviewer #1 is negligibly small, which is confirmed by the absence of twofold oscillations above T^* . The mechanical rotator is situated at the top of the cryostat, so it is always at room temperature. The misalignment also has been checked

in the superconducting state at low temperatures (see Fig.S3). Thus the observed temperature dependence cannot be explained by the misalignment effect."

As discussed above, the validity of their interpretation requires alignment of three planes (i),(ii),(iii) -- adjusting vector magnet at one temperature only achieves cancellation (at one temperature) between wobbling of plane (i) with respect to (ii) and plane (ii) with respect to (iii). This cancellation is destroyed as soon as temperature changes. This concern is even more severe because the magnitude of the expected signal due to wobbling is about 1/10th of the change in $\Delta\chi_{\perp} = \chi_{cc} - \chi_{aa}$, -- close to the observed magnitude. Here 0.1 is the angle of misalignment of the sample on the lever (plane (i) with respect to plane (ii)) in radians. In this context, the measured signal is direct consequence of the temperature behavior of $\chi_c - \chi_a$ rather than in-plane anisotropy $\chi_{bb} - \chi_{cc}$

(Our reply #2-6)

As we have commented in this reply, the misalignment of the sample with respect to the magnetic field cannot explain our experimental observation. Both the amplitude and the phase due to misalignment are inconsistent with the experimental observations. The experimental procedures are as follows.

1. We first determine the alignment of the ab-plane at 80 K in the superconducting state.
2. Then using the vector magnetic field and mechanical rotator, we measure in-plane magnetic torque as a function of azimuth angle ϕ , with the magnetic field exactly applied within the ab-plane.
3. Measurements of the $\tau(\phi)$ curves are repeated for all the temperatures in Figs.4. By analyzing all the two-fold oscillations, we obtain the temperature dependences of χ_{ab} and $\chi_{aa} - \chi_{bb}$.
4. Consequently, we find that finite χ_{ab} emerges below T^* . As we have commented above, the alignment of the sample is double checked after completing the experiments.

To explain the details of experimental set-up for non-expert readers, we have revised the Methods section and Supplementary Information.

Reviewers' comments:

Reviewer #3 (Remarks to the Author):

The authors look at the ab-plane anisotropy in the tetragonal system Hg cuprate. With torque magnetometry, they observe a two-fold anisotropy upon cooling below the pseudogap temperature, which is taken as evidence for a nematic phase transition similar to that observed in the pnictides and in YBCO. However, the suppression of the two-fold anisotropy near the CDW transition in Hg1201 and its orientation along the diagonal of the Cu-O plane suggests that nematicity and the CDW compete with one another. The fact that the CDW modulation and the nematicity in other cuprates both occur along the bond directions makes it more difficult to distinguish whether nematicity is a precursor of the CDW or not. For this reason, these results on the tetragonal cuprate are of significantly high interest and would have broad impact. The paper is also very well written, enjoyable to read, and the data is presented clearly, but the discussion of the data, is in my opinion, not very clear.

This is very challenging work experimentally because there are many systematic errors that come into play. Unfortunately, their claim of detecting purely ab-plane anisotropy necessitates even closer attention to detail regarding these systematic errors than usual. With the present manuscript, I can't say with confidence that their measured two-fold anisotropy is entirely due to the ab-plane anisotropy, but this may be due to a lack of clarity in their discussion.

My understanding is that the authors go to several specific phi positions and rotate the magnetic field in theta (in the superconducting state) to determine precisely the position of the ab-plane. I have to assume that this is all done without removing the sample from the lever (only confused because fig. S2 suggests rotation is in phi, while fig. S3 is discussing rotation in theta). Is the rotation in theta controlled by the vector magnet so that the sample is not unmounted, lever rotated, etc between measurements in S3 and figure 3 of the main text for example? If so, please make this clear in the manuscript. Otherwise, under continual rotation in phi (as shown in figure S2), misalignment between the sample and rotation stage will mean that the ab-plane is not always in the H plane (red plane in S2b) determined by the theta scans at multiple phis. If the phi and theta scans are done without sample realignment, then misalignment between the sample and the rotation stage plane is accounted for due to rotation of the magnetic field for several phis, and thus the ab-plane is determined? In this case, the authors believe H to always be applied within 0.1 degree of the ab-plane. If I understand this correctly, it is important to clarify this discussion in order to assist readers in understanding how one misalignment issue is taken into account. Generally speaking, the authors comment in many places about "misalignment between the sample and field/cantilever", using "field/cantilever" as if they are the same thing, but these are very different and each must be dealt with independently, which brings me to the next point.

While the authors seem to take great care in orienting the magnetic field with respect to the crystallographic directions, the present version of the manuscript does not address misalignment between the sample and the vibration plane of the cantilever. In the methods section, the authors acknowledge that there is an unavoidable misalignment between the lever and the sample plane, which leads to oscillations from the out-of-plane anisotropy. But then it's unclear to me how the discussion that follows helps to convince the reader that this is not the reason for the two-fold oscillations that are taken as being due to ab-plane anisotropy. In particular, I don't understand the statement "we calculate the expected torque response, for the following cases assuming that the misalignment of the sample with respect to the lever is as large as the apparent misalignment found at 80 K". The part in parenthesis describing the angles comes about from determination of magnetic field w.r.t. to crystal axes and has nothing to do with the angle misalignment with the lever. In practice, the misalignment between the crystal axes and the lever vibration plane could easily be 10 degrees (as it comes from both rotation of the sample flat on the lever and any tilt up or down from the flat lever surface). I believe these are the details that reviewer #2 was requesting, and I too would find them particularly useful. In the quoted statement above, do the

authors mean that they somehow incorporate in their simulations a misalignment that is also between the sample w.r.t. to the lever vibration plane? This is what is relevant and not really discussed in detail or if it is, it is unclear. Again, this is really confusing because in all that follows it's described as "misalignments between the sample and the field/cantilever". When I read about the specific cases, it seems that only misalignment between the crystal axes and the plane that the magnetic field rotates through is included (and I'm not sure why because the authors already convinced me that this should be really small...).

An important misalignment issue (between the crystal axes and the vibration plane of the lever) appears to not be considered in this work. This misalignment could lead to a response that is much more susceptible to the out-of-plane susceptibility (because there is still some component of magnetic field along c , albeit small if I trust the "computer-controlled vector magnet and mechanical rotator") because ab-plane anisotropy is not probed exclusively (i.e., the ab-plane is not in the plane that is the natural bending mode of the cantilever). Unfortunately, the out-of-plane anisotropy onsets at the same temperature (between 200 and 250 K in Fig S5) that is taken as the onset of the ab-plane anisotropy in this work. Furthermore, the contribution of the out-of-plane susceptibility (due to a tilt of the ab-plane from the lever vibration plane) can be estimated as the tilt angle squared multiplied by the out-of-plane susceptibility and if I assume a very small angle of just 5 degrees, I find that the out-of-plane contribution is 38% of the measured "ab-plane" signal at 100K ($p=0.11$ doping). 5 degrees may be an underestimate, as I find in practice that a 5 degree misalignment between the crystal axes and the vibration plane can be very difficult to achieve.

Two very minor comments regarding the manuscript:

1) Panel d is not described in Figure 4.

2) In the first paragraph on page 6, starting "Deep inside the nematic phase.." About midway through this paragraph, there is a sentence which starts with "These temperatures.." and because you talking about T_c in YBCO just prior, it's important to make it clear that you are now switching back to discussing the suppression temperatures observed in Hg1201.

In summary, this is a nice and impactful result and I believe the authors try to do their due diligence in removing the systematic errors, and so I hope I can be convinced that the misalignment between the crystal axes and the vibration plane is not the reason for the two-fold oscillations. I certainly don't think this misalignment can be completely disregarded as is in the current text (e.g., in the discussion surrounding "In case (B),.." on page 15).

Reviewer #3 (Remarks to the Author):

The authors look at the ab-plane anisotropy in the tetragonal system Hg cuprate. With torque magnetometry, they observe a two-fold anisotropy upon cooling below the pseudogap temperature, which is taken as evidence for a nematic phase transition similar to that observed in the pnictides and in YBCO. However, the suppression of the two-fold anisotropy near the CDW transition in Hg1201 and its orientation along the diagonal of the Cu-O plane suggests that nematicity and the CDW compete with one another. The fact that the CDW modulation and the nematicity in other cuprates both occur along the bond directions makes it more difficult to distinguish whether nematicity is a precursor of the CDW or not. For this reason, these results on the tetragonal cuprate are of significantly high interest and would have broad impact. The paper is also very well written, enjoyable to read, and the data is presented clearly, but the discussion of the data, is in my opinion, not very clear.

This is very challenging work experimentally because there are many systematic errors that come into play. Unfortunately, their claim of detecting purely ab-plane anisotropy necessitates even closer attention to detail regarding these systematic errors than usual. With the present manuscript, I can't say with confidence that their measured two-fold anisotropy is entirely due to the ab-plane anisotropy, but this may be due to a lack of clarity in their discussion.

First of all, we thank Reviewer #3 for taking her/his time to review our manuscript. We also appreciate her/his high evaluation on our results of "diagonal" nematicity in tetragonal cuprates, which would be significantly of high interest and have broad impact. Having read the comments by Reviewer #3, we realized that, indeed, our discussion was unnecessarily complicated. Accordingly, we have revised the manuscript, in particular regarding the experimental procedures and the discussion of the misalignment issues. In fact, all the major comments by Reviewer #3 basically boil down to the question of sample misalignment w.r.t. the cantilever. To address the concerns raised by Reviewer #3, we have performed additional experiments and analysis. In this reply, we clearly demonstrate that

- The observed in-plane two-fold oscillations arise neither from misalignment of the sample w.r.t. the lever nor from misalignment of the field w.r.t. the sample.
- New measurements with non-zero out-of-plane field component (see Figs. R3 and R4) clearly reveal the emergence of the diagonal nematicity at $T < T^*$.

We believe our detailed response shown below will remove all the concerns raised by

Reviewer #3.

My understanding is that the authors go to several specific phi positions and rotate the magnetic field in theta (in the superconducting state) to determine precisely the position of the ab-plane. I have to assume that this is all done without removing the sample from the lever (only confused because fig. S2 suggests rotation is in phi, while fig. S3 is discussing rotation in theta). Is the rotation in theta controlled by the vector magnet so that the sample is not unmounted, lever rotated, etc between measurements in S3 and figure 3 of the main text for example? If so, please make this clear in the manuscript. Otherwise, under continual rotation in phi (as shown in figure S2), misalignment between the sample and rotation stage will mean that the ab-plane is not always in the H plane (red plane in S2b) determined by the theta scans at multiple phis. If the phi and theta scans are done without sample realignment, then misalignment between the sample and the rotation stage plane is accounted for due to rotation of the magnetic field for several phis, and thus the ab-plane is determined? In this case, the authors believe H to always be applied within 0.1 degree of the ab-plane. If I understand this correctly, it is important to clarify this discussion in order to assist readers in understanding how one misalignment issue is taken into account.

Before going into the details of the response, let us make our experimental situations clearer. It should be noted that there are basically two coordinate systems used in our experiments. One is the XYZ-system based on the rotation mechanism (Fig. R1), which we use to determine the position of the sample plane. The other is the a,b,c-axes of the sample (Fig. R2). The presence of the multiple axes was perhaps the most confusing issue with the previous version of our manuscript. Indeed, now we use uppercase letters for the polar (Θ) and azimuthal (Φ) angles for the former coordinate, and lowercase (θ and ϕ) for the latter to avoid confusion.

Fig. R1

Fig. R2

In our experiments, as Reviewer #3 correctly comments, we first measure $\tau(\Theta)$ curves at

various Φ (Fig. S3a), and determine the position of the crystal ab-plane ($\Delta\Theta_m$) as a function of Φ (Fig. S3b). The misalignment $\Delta\Theta_m$ from the XY-plane is eliminated at each Φ by using the 2D vector magnet. Then, without changing the setup or removing the sample, we measure the in-plane magnetic torque with a field alignment better than 0.1 deg. We now explicitly discuss this procedure, and have revised the corresponding figures in the updated manuscript.

Generally speaking, the authors comment in many places about “misalignment between the sample and field/cantilever”, using “field/cantilever” as if they are the same thing, but these are very different and each must be dealt with independently, which brings me to the next point.

While the authors seem to take great care in orienting the magnetic field with respect to the crystallographic directions, the present version of the manuscript does not address misalignment between the sample and the vibration plane of the cantilever. In the methods section, the authors acknowledge that there is an unavoidable misalignment between the lever and the sample plane, which leads to oscillations from the out-of-plane anisotropy. But then it’s unclear to me how the discussion that follows helps to convince the reader that this is not the reason for the two-fold oscillations that are taken as being due to ab-plane anisotropy.

As Reviewer #3 correctly points out, the misalignment of the magnetic field w.r.t. the sample (field misalignment) is independent of that of the sample w.r.t. the lever (mount misalignment). We however stress that the in-plane anisotropy can be accurately measured under the aligned field condition ($H_c = 0$), independent of the mount misalignment. As we described in the manuscript, the signal detected by the cantilever is $\tau^{\text{lever}} = \boldsymbol{\tau} \cdot \mathbf{e}_z = \tau_a \mathbf{e}_a + \tau_b \mathbf{e}_b + \tau_c \mathbf{e}_c$, where $\mathbf{e}_z = (\mathbf{e}_a, \mathbf{e}_b, \mathbf{e}_c)$ is the normal unit vector of the bending plane (see, Fig.R2) and

$$\begin{aligned}\tau_a &= \mu_0 V [\chi_{ab} H_a H_c - (\chi_{cc} - \chi_{bb}) H_b H_c] \\ \tau_b &= \mu_0 V [\chi_{ab} H_b H_c - (\chi_{cc} - \chi_{aa}) H_a H_c] \\ \tau_c &= \mu_0 V [\chi_{ab} (H_b^2 - H_a^2) + (\chi_{aa} - \chi_{bb}) H_a H_b] .\end{aligned}$$

Thus, only $\tau^{\text{lever}} = \boldsymbol{\tau} \cdot \mathbf{e}_z = \tau_c \mathbf{e}_c$ survives when $H_c = 0$, even if mount misalignment is present. We explicitly comment on these points in the revised manuscript.

In particular, I don’t understand the statement “we calculate the expected torque response, for the following cases assuming that the misalignment of the sample with respect to the

lever is as large as the apparent misalignment found at 80 K". The part in parenthesis describing the angles comes about from determination of magnetic field w.r.t. to crystal axes and has nothing to do with the angle misalignment with the lever.

What we determine in our experiments is the alignment of the sample from the basal XY-plane of the rotation mechanism. Since the bending plane of the lever is essentially in the XY-plane of the rotating stage, the sample alignment from the XY-plane can be regarded as the mount misalignment, at least to the first approximation. In reality, however, there also exists a small misalignment of the lever w.r.t. the XY-plane of the rotating stage, which cannot be accurately determined. Therefore, we refer to an "apparent misalignment" and use it in our calculation. It should be noted that the difference between the bending plane of the lever and the XY-plane of the rotation stage only results in a slight modification of \mathbf{e}_z and does not alter the above discussion when the magnetic field is correctly applied within the ab-plane.

In practice, the misalignment between the crystal axes and the lever vibration plane could easily be 10 degrees (as it comes from both rotation of the sample flat on the lever and any tilt up or down from the flat lever surface). I believe these are the details that reviewer #2 was requesting, and I too would find them particularly useful.

As we show in Fig. S3b, mount misalignment can be ~10 deg depending on Φ . However, as discussed above, the signal is not contaminated by the out-of-plane component as long as the magnetic field is applied within the ab-plane.

In the quoted statement above, do the authors mean that they somehow incorporate in their simulations a misalignment that is also between the sample w.r.t. to the lever vibration plane? This is what is relevant and not really discussed in detail or if it is, it is unclear. Again, this is really confusing because in all that follows it's described as "misalignments between the sample and the field/cantilever". When I read about the specific cases, it seems that only misalignment between the crystal axes and the plane that the magnetic field rotates through is included (and I'm not sure why because the authors already convinced me that this should be really small...).

For the case (B) shown in Fig. S6e and f, we calculate the expected torque when field misalignment is present for arbitrary (θ, ϕ) but mount misalignment is absent. In this case, only $\tau^{\text{lever}} = \boldsymbol{\tau} \cdot \mathbf{e}_z = \tau_c e_c$ is measured because $\mathbf{e}_z = (0, 0, e_c)$ regardless of the field

misalignment. By contrast, for the case (C) shown in Fig. S6g and h, we calculate the expected torque when both the mount misalignment and the field misalignment are present. In the revised manuscript, we separately describe the issues of field and mount misalignments and clarify the conditions of our calculations.

An important misalignment issue (between the crystal axes and the vibration plane of the lever) appears to not be considered in this work. This misalignment could lead to a response that is much more susceptible to the out-of-plane susceptibility (because there is still some component of magnetic field along c , albeit small if I trust the “computer-controlled vector magnet and mechanical rotator”) because ab-plane anisotropy is not probed exclusively (i.e., the ab-plane is not in the plane that is the natural bending mode of the cantilever).

Please see our response above. In contrast to Reviewer #3's comments, our calculation for case (C) includes both field misalignment and mount misalignment. Then, the expected torque is mapped in Fig. S6g as a function of field misalignment (θ, ϕ). When magnetic field is applied within the ab-plane, only the in-plane anisotropy is measured even if the sample is mounted with non-zero misalignment w.r.t. the lever.

Unfortunately, the out-of- plane anisotropy onsets at the same temperature (between 200 and 250 K in Fig S5) that is taken as the onset of the ab-plane anisotropy in this work. Furthermore, the contribution of the out- of-plane susceptibility (due to a tilt of the ab-plane from the lever vibration plane) can be estimated as the tilt angle squared multiplied by the out-of-plane susceptibility and if I assume a very small angle of just 5 degrees, I find that the out-of-plane contribution is 38% of the measured “ab-plane” signal at 100K ($p=0.11$ doping). 5 degrees may be an underestimate, as I find in practice that a 5 degree misalignment between the crystal axes and the vibration plane can be very difficult to achieve.

We respectfully disagree with these comments. As we have noted so far, mixing of the out-of-plane components occurs only when both mount misalignment and field misalignment are present, and as Reviewer #3 acknowledges, we can accurately eliminate the field misalignment by vector magnet. Therefore, only the in-plane anisotropy is measured. Even if we hypothetically assume that the field is applied with non-zero c -axis component H_c , this cannot be the reason for the onset of “diagonal” nematicity, because in such a case, the phase of the observed two-fold oscillations should be randomly oriented. By contrast, the phase of the observed two-fold oscillations is always fixed as $\tau_{2\phi} \sim \cos 2\phi$.

To further demonstrate the validity of our experiments, and that field misalignment is not the cause of the diagonal nematicity, we performed new magnetic torque measurements with a non-zero out-of-plane field component. Here, the sample is remounted onto the lever and an apparent misalignment from the XY plane is determined as $\Theta_0 = 10.6$ deg and $\Phi_0 = 274$ deg by $\tau(\Theta)$ measurements.

Fig. R3

Fig. R4

Figures R3a and b show the magnetic torque for $p \sim 0.125$ recorded under conical field rotations at several θ for $T = 240$ K and 160 K, respectively. In Figs. R4a-c, we map the results of the magnetic torque in the (θ, ϕ) plane for $T = 240, 180,$ and 160 K, respectively. Figures R4d-f depict the expected torque amplitude when both field and mount

misalignments are present. In Fig. R4g, we plot the expected torque, in case we misidentify the direction of the sample plane, such that the magnetic field is rotated based on a different plane of, e.g., $\theta' = (\theta - 5^\circ)\cos(\phi - 20^\circ)$.

- At $T = 240 \text{ K}$ ($> T^*$), in-plane anisotropy is absent and the torque amplitude shows symmetric behaviour in the (θ, ϕ) plane. This indicates that $\theta = 90 \text{ deg}$ correctly captures the direction of the ab-plane of the crystal. Otherwise, the response of the torque would be distorted as in Fig. R4g.
- The emergence of the in-plane anisotropy below T^* is clearly seen at $\theta = 90 \text{ deg}$. As a result, the torque response becomes asymmetric in the (θ, ϕ) plane, because two-fold oscillations of the diagonal nematicity are mixed into the signal.
- This deformation of the torque amplitude in the (θ, ϕ) plane below T^* is essentially different from the simple distortion expected for the case when the sample plane is misidentified.
- Our results under non-zero H_c component show excellent agreement with the expected torque response.

These new results clearly demonstrate that neither field misalignment nor mount misalignment is the reason for the observed onset of the two-fold oscillations that reveal the emergence of diagonal nematicity below T^* . We have included these new experiments in the Methods section and Fig. S8 and Fig. S9 in the revised manuscript.

Two very minor comments regarding the manuscript:

- 1) Panel d is not described in Figure 4.
- 2) In the first paragraph on page 6, starting “Deep inside the nematic phase..” About midway through this paragraph, there is a sentence which starts with “These temperatures..” and because you talking about T_c in YBCO just prior, it’s important to make it clear that you are now switching back to discussing the suppression temperatures observed in Hg1201.

Thank you very much for reading the manuscript carefully. We have made revisions on these minor points.

In summary, this is a nice and impactful result and I believe the authors try to do their due diligence in removing the systematic errors, and so I hope I can be convinced that the misalignment between the crystal axes and the vibration plane is not the reason for the two-fold oscillations. I certainly don’t think this misalignment can be completely disregarded

as is in the current text (e.g., in the discussion surrounding “In case (B),...” on page 15).

We believe that we have clearly demonstrated that the misalignment between the crystal axes w.r.t. the vibration plane of the lever is not the reason for the two-fold oscillations of the diagonal nematicity. With our reply and the revised manuscript, we hope that Reviewer #3 will now recommend our manuscript for publication in *Nature Communications*.

REVIEWERS' COMMENTS:

Reviewer #3 (Remarks to the Author):

Thank you to the authors for addressing the major concern that I had from the previous round of review - namely that misalignment between the vibration plane of the lever and the plane of rotation had not been carefully considered. They clarified discussion throughout the text to avoid confusion for future readers, performed new measurements, and did calculations for the expected torque assuming reasonable misalignments. I think all of these efforts help to strengthen and substantiate their claim that the observed response is due to anisotropy within the CuO₂ plane. I support publication of this manuscript.